# MathArena: Evaluating LLMs on Uncontaminated Math Competitions

**Mislav Balunović**[1,2], **Jasper Dekoninck**[1], **Ivo Petrov**[2], **Nikola Jovanović**[1], **Martin Vechev**[1,2]
[1]ETH Zurich, [2]INSAIT, Sofia University
{mislav.balunovic,jasper.dekoninck,nikola.jovanovic,martin.vechev}@inf.ethz.ch,
ivo.petrov@insait.ai

🌐 https://matharena.ai/
 https://github.com/eth-sri/matharena

## Abstract

The rapid advancement of reasoning capabilities in large language models (LLMs) has led to notable improvements on mathematical benchmarks. However, many of the most commonly used evaluation datasets (e.g., AIME 2024) are widely available online, making it difficult to disentangle genuine reasoning from potential memorization. Furthermore, these benchmarks do not evaluate proof-writing capabilities, which are crucial for many mathematical tasks. To address this, we introduce MATHARENA, a new benchmark based on the following key insight: recurring math competitions provide a stream of high-quality, challenging problems that can be used for real-time evaluation of LLMs. By evaluating models as soon as new problems are released, we effectively eliminate the risk of contamination. Using this framework, we find strong signs of contamination in AIME 2024. Nonetheless, evaluations on harder competitions, such as CMIMC 2025, demonstrate impressive reasoning capabilities in top-performing models. MATHARENA is also the first benchmark for proof-writing capabilities. On IMO 2025, top models achieve slightly less than 40%, demonstrating both notable progress and significant room for improvement. So far, we have evaluated over 50 models across seven competitions, totaling 162 problems. As an evolving benchmark, MATHARENA will continue to track the progress of LLMs on newly released competitions, ensuring rigorous and up-to-date evaluation of mathematical reasoning.

## 1 Introduction

Recent advances in the mathematical reasoning capabilities of large language models (LLMs) [20, 7] have raised the following three concerns about the adequacy of existing mathematics benchmarks:

**1. Contamination risks**: Many benchmarks are sourced from publicly available math competitions, which are accessible online and often used to train LLMs. This leaves them susceptible to data contamination, making it difficult to measure progress accurately. Data contamination can occur either through indirect inclusion of benchmark problems in training data or by using benchmark performance for hyperparameter tuning or model selection. For instance, we find that the popular AIME 2024 dataset is significantly contaminated by most leading LLMs, making the benchmark unsuitable for evaluating the models' capabilities.

**2. High-cost, private benchmarks**: To mitigate contamination, some leading benchmarks—such as FrontierMath [15] and HLE [29]—adopt a private, human-curated approach. While effective in avoiding data leakage, these datasets pose several major issues. First, their private nature raises concerns about reproducibility and transparency, making it impossible to verify the results accurately.

39th Conference on Neural Information Processing Systems (NeurIPS 2025) Track on Datasets and Benchmarks.

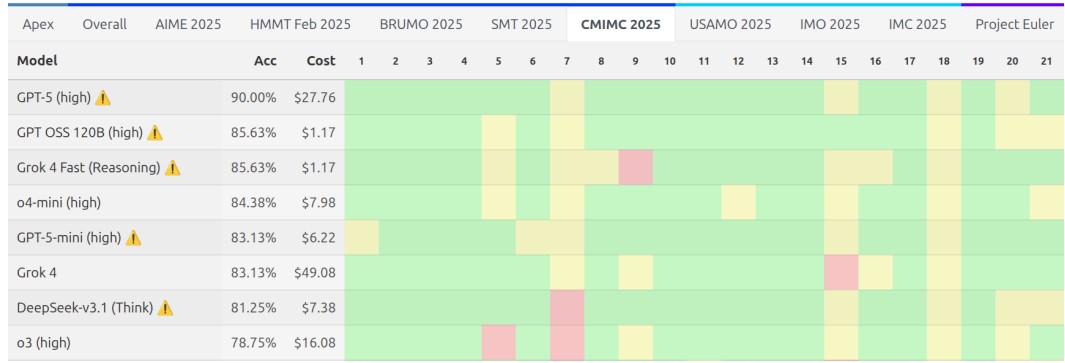

Figure 1: MATHARENA leaderboard. The picture shows the CMIMC competition held in March 2025. Cell color denotes the pass rate of the model on the problem out of 4 attempts. Warning signs show a possible contamination risk due to the model being released after the competition.

Moreover, the benchmark creators may selectively grant access to certain organizations [10], creating an uneven playing field. And finally, the high cost of developing such datasets is prohibitive. For instance, HLE required a $500,000 prize pool to incentivize contributions.

**3. Emphasis on final answers**: Most existing benchmarks, including HLE and FrontierMath, primarily evaluate problems with single final answers. This can be misleading, as models may arrive at the correct answer through pattern recognition or brute-force enumeration, rather than genuine mathematical reasoning. Such benchmarks fall short of capturing the depth and rigor of problems found in mathematical olympiads, which often require detailed proofs and multi-step logic. Furthermore, most practical applications of LLMs in mathematics involve generating proofs or explanations, rather than simply providing final answers.

**MATHARENA: A new benchmark for mathematical reasoning**    We introduce MATHARENA, a dynamic publicly available benchmark which addresses the above limitations by evaluating on newly released math competitions (see Fig. 1). Our core insight is that recurring math competitions produce a rich source of high-quality, uncontaminated problems. These problems are pre-vetted by the competition organizers for originality, ensuring that similar problems have not appeared previously and thereby reducing contamination risk. By evaluating models on competitions occurring after model release, MATHARENA eliminates the risk of contamination and offers a clean, forward-looking measure of progress. Furthermore, some of the included competitions (e.g., IMO 2025) have proof-based problems that are absent in other benchmarks. Unlike private or static benchmarks, MATHARENA is fully transparent, reproducible, and continuously updated throughout the year as new problems become available. This enables continuous adaptation to the evolving landscape of mathematical reasoning, ensuring that the included competitions remain relevant and challenging. We implement the entire MATHARENA pipeline for parsing, solving, and verifying problem solutions and release the code, data, and model responses as open source.

So far, we have evaluated over 50 models across seven competitions, totaling 162 problems. Our results indicate that GPT-5, GROK 4, and GEMINI-2.5-PRO are the top-performing models on the included competitions, outperforming the top 1% of human participants. However, we also find room for improvement on proof-based competitions, with models scoring below 40% on the IMO 2025. This highlights the need for further research in this area.

**Key contributions**    In summary, our key contributions are as follows:

- We introduce MATHARENA, a benchmark that leverages newly released competitions to evaluate LLMs, eliminating contamination while being fully transparent and reproducible.
- We propose a scalable evaluation pipeline for parsing, solving, and verifying problems from diverse competition formats, including final-answer and proof-based problems.
- We compare and thoroughly analyze the performance of state-of-the-art models on these competitions, highlighting significant progress made within the past year.

## 2 Related Work

In this section, we discuss the key prior approaches for evaluating mathematical reasoning.

**Public answer-based benchmarks**   The most widely used benchmarks evaluate models by comparing their outputs to fixed ground-truth answers—typically numerical values or closed-form expressions. Early benchmarks such as GSM8K [6] and MATH [17] have largely been saturated by recent language models. Even more challenging competitions, like AIME 2024, have seen similar progress and are close to saturation. Omni-MATH [14], OlympiadBench [16], HARP [36], and OlymMATH [30] increase difficulty by incorporating harder problems from olympiad competitions. However, sourcing problems from past competitions, which have been available online for years, makes it difficult to track progress due to data contamination risks. This concern is supported by evidence in the case of GSM8K [37], and we confirm contamination of AIME 2024 in Section 4.

**Private answer-based benchmarks**   FrontierMath [15] is a recently introduced private benchmark designed to be significantly more challenging, with problems that demand mathematical reasoning combined with a deep background in research-level mathematics. Similarly, Humanity's Last Exam [29] has collected a large number of private challenging problems across dozens of subjects. While their extreme difficulty makes an interesting target for frontier models, the private nature of these benchmarks makes standardized evaluation and fair model comparison difficult. Furthermore, this difficulty level makes tracking progress challenging, particularly for open source models and models on the Pareto frontier of cost-performance. Finally, the private nature of the benchmark raises concerns about reproducibility and transparency, as access to these benchmarks has been selectively granted to certain organizations [10].

**Proof-based benchmarks**   Another line of evaluation focuses on verifying the correctness of reasoning traces rather than final answers. A common strategy is to require LLMs to generate formal proofs in systems like Lean, Coq, or Isabelle, enabling automatic verification. Datasets and benchmarks in this category include miniF2F [39], FIMO [21], PutnamBench [32], and LeanWorkbook [35]. However, these approaches often underutilize the natural language capabilities of LLMs and are limited by the models' ability to produce correct formal code. Concurrent work [22] reveals that models typically fail to generate fully rigorous proofs in natural language. Even for the correctly solved problems, the inclusion of the IMO shortlist problems likely leads to significant contamination, and the size of the benchmarks makes it infeasible to evaluate new models on all problems. GHOSTS [13] manually evaluate the proof-writing capabilities of GPT-4, but their benchmark is limited to just two older models and has not been updated since 2023.

**Dynamic benchmarks**   To address contamination and adapt to evolving capabilities, some benchmarks are designed to be continuously updated with new problems. LiveBench [33], for instance, evaluates LLMs across domains including coding, data analysis, and mathematics. The mathematics portion in particular includes slightly harder than MATH-level problems, as well as fill-in-the-blank proof-based tasks, making it easier than MATHARENA, while also not evaluating rigorous proving capabilities. Another similar work to ours is LiveAoPSBench [23], which allows evaluating models on a snapshot of problems from a particular point in time. This can be seen as a retroactive simulation of live evaluation as performed in MATHARENA. However, the benchmark is not updated and does not contain problems from 2025, which precludes the evaluation of recent frontier models.

**Perturbation-based benchmarks**   Another way to mitigate contamination risks is to generate new problems by perturbing existing ones [19, 24, 40]. While this strategy reduces overlap, it does not fully eliminate contamination: perturbed problems rely on the same underlying reasoning patterns. In contrast, our approach introduces entirely new problems that require new high-level reasoning strategies.

**Other benchmarks**   Finally, some benchmarks adopt less conventional methods to evaluate mathematical reasoning. For example, MathTrap [38] evaluates logical consistency in model responses, while MathConstruct [8] focuses on problems that require constructive proofs. These approaches provide a more diverse view of the mathematical reasoning capabilities of the models. However, these benchmarks typically require expensive human data curation, which limits scalable evaluation.

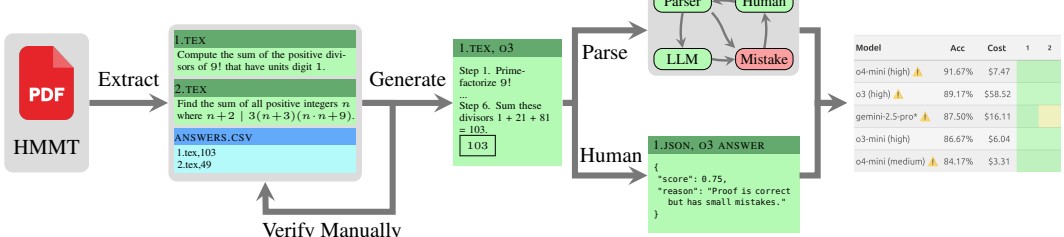

Figure 2: The pipeline for constructing MATHARENA. When a new competition is released, we first extract the problems and answers. We then query all selected models to obtain their responses. Depending on the problem type, we either use an automated parser or human graders for evaluation. Finally, we report scores on a public leaderboard with a GUI for viewing individual model answers

# 3 MATHARENA

In this section, we describe the pipeline used to construct MATHARENA, as shown in Fig. 2. The process begins by selecting a sufficiently challenging and reputable competition and extracting its problems and solutions (Section 3.1). Next, we evaluate a selected set of models on these problems, ensuring a fair comparison and avoiding data leakage (Section 3.2). Depending on the type of problem, either final-answer or proof-based, we use different methods for parsing and evaluation (Section 3.3). For final-answer problems, we use an automated rule-based parser to extract answers. For proof-based problems, human graders evaluate the model outputs. Finally, we compute leaderboard rankings and perform statistical post-processing to ensure accuracy and reliability (Section 3.4).

## 3.1 Competition Selection and Extraction

**Competition selection** To effectively repurpose high-quality math competitions for LLM evaluation, we carefully select which competitions to include in MATHARENA and ensure accurate formatting of each problem. Table 1 shows a calendar of competitions currently included in MATHARENA, along with additional competitions we plan to incorporate. At present, MATHARENA includes seven competitions comprising a total of 162 problems. We categorize competitions based on whether they consist of final-answer or proof-based problems. Final-

Table 1: Calendar of completed and planned competitions. $N$ denotes the number of problems.

| Competition | Type | Date | N | Current |
|---|---|---|---|---|
| AIME | Answer | Feb | 30 | ✓ |
| HMMT FEB. | Answer | Feb | 30 | ✓ |
| USAMO | Proof | Mar | 6 | ✓ |
| CMIMC | Answer | May | 40 | ✓ |
| BRUMO | Answer | Apr | 30 | ✓ |
| IMO | Proof | Jul | 6 | ✓ |
| PROJECT EULER | Answer | - | 20+ | ✓ |
| MMATHS | Answer | Nov | TBD | ✗ |
| DMM | Answer | Nov | TBD | ✗ |
| PUMAC | Answer | Nov | TBD | ✗ |
| PUTNAM | Proof | Dec | 12 | ✗ |

answer competitions are easier to evaluate but tend to be less challenging. For these, we focus on high-difficulty competitions such as AIME (a qualifier for the USAMO) and several more difficult university-organized tournaments. We experimented with other well-known competitions, such as Kangaroo, and excluded them as they are already saturated by existing models.

Proof-based competitions pose a greater challenge and are more representative of deep mathematical reasoning. However, they also require manual evaluation, as scalable automated grading of proofs remains an open problem. To ensure high evaluation quality, we use human graders to evaluate proofs and focus on a small set of core competitions: USAMO (US high-school olympiad), IMO (International Math Olympiad), and the Putnam competition (US undergraduate level).

In addition to the standard mathematical competitions, we include problems from Project Euler [11], a popular online platform that emphasizes mathematical problem solving through code implementations. Unlike traditional competitions, Project Euler does not follow a fixed schedule or problem set. Instead, it maintains a continually expanding collection of problems. For evaluation, we focus only on the most recent problems and plan to update this subset regularly as new ones are released.

**Problem extraction** After selecting competitions, we extract the problems from their original sources and format them into a standardized template. We manually verify each problem for typographical errors, inconsistencies, or formatting issues.

## 3.2 Model Selection and Solution Generation

**Model selection**    MATHARENA is continuously updated with newly released models. To avoid an overly cluttered leaderboard, we only select models that meet at least one of the following criteria: (i) the model competes for the top score in a given competition (e.g., GPT-5, GEMINI-2.5-PRO, GROK 4), (ii) the model competes for the top-performing open-weight option (e.g., DEEPSEEK-R1, QWEN3), or (iii) the model competes for a Pareto-optimal point on the cost-performance tradeoff curve (e.g., GROK 4 FAST, GPT-OSS-20B). We exclude non-reasoning models, as they consistently underperform reasoning models and do not satisfy any of the selection criteria.

**Solution generation**    Each model is evaluated once per competition using the hyperparameters recommended by the model providers, without further tuning. This avoids overfitting and reduces the risk of information leakage. For answer-based competitions, we prompt the models to output their answer inside of a boxed environment, while for proof-based competitions, we prompt the models to output the entire proof. In App. D, we provide the prompts used for each competition. To account for stochasticity, each model generates four responses per question, and we report the average score across these runs. Models are evaluated close to the competition date, minimizing contamination risk. If a model was released after the competition date, this is clearly indicated on the leaderboard. Examples of model outputs and questions are shown in App. E.

**Project Euler tools**    For Project Euler, we allow models to use tools to execute code, as this is often necessary to solve the problems. We provide a Python and C++ interpreter for this purpose. Models can generate code snippets that are executed in a secure sandbox environment, and the output can be used in subsequent reasoning steps. We limit the number of code executions to 20 per problem.

## 3.3 Solution Grading

Our grading strategy differs significantly between final-answer and proof-based problems. We outline details of both approaches below. These approaches are depicted in Fig. 2 with *Parse* (answer-based) and *Human* (proof-based) branches.

**Answer-based competitions**    Answer-based competitions typically allow fairly accurate automated grading by extracting the final answer from boxed and using rule-based parsing on the extracted string. However, given the small size of these competitions, *fairly* accurate parsing is not good enough, as even minor parser errors can have a disproportionate impact. To this end, we develop a custom rule-based parser that converts arbitrary LaTeX strings into structured sympy expressions, capable of handling complex mathematical objects such as fractions, lists, and radicals. These expressions are then checked for equivalence with the ground truth answer using sympy. Since model outputs often vary in formatting, parser robustness is crucial. We implement two measures to ensure correctness.

First, we developed a GUI to support manual review of model answers, highlighting: (i) suspiciously short outputs, which may indicate truncation due to token limits, (ii) parser errors, and (iii) instances where the correct answer appears in the reasoning trace but is not successfully extracted. In the first case, if a model frequently exhibits this issue, we may consider re-running it with a different API provider, as the used provider likely limits the number of tokens per generation. In the other cases, we perform manual verification of all such flagged problems. Second, we incorporate an LLM-based judge, using the GEMINI-2.5-FLASH model, which evaluates whether the model's final answer is semantically equivalent to the ground truth. If the parser and judge disagree, we manually inspect the model response and update the parser as needed.

**Proof-based competitions**    Automated grading is currently insufficient for proof-based problems, so we rely on expert human graders for precise grading. First, as competitions typically do not publish their grading scheme, expert graders develop a structured grading scheme meant to closely resemble the one used at the actual competition, e.g., rewarding points for partial progress. Next, graders receive anonymized solutions from the selected models and grade them according to the previously developed scheme. Two independent judges grade each solution, providing not only a final score but also a justification for their decision. We refer readers to [28] for further details of the procedure.

## 3.4 Leaderboard and Post-Processing

Once model outputs have been evaluated, we perform several post-processing steps to ensure the reliability of reported results. These include leaderboard construction and statistical variance estimation.

Table 2: The results of our numerical answer evaluation on the latest models evaluated on all competitions. Measured cost is the average cost to run a model one time on a single competition, and accuracy is the average accuracy across all 4 competitions. Green cells denote that the model was released after the competition date. Human performance is reported for the top $1\%$ of participants in the AIME and HMMT competitions. For BRUMO and CMIMC, the human performance is not available.

| Model | AIME | HMMT | BRUMO | CMIMC | Acc (avg) | Cost (avg) |
|---|---|---|---|---|---|---|
| GPT-5 (HIGH) | 95.0 | 88.3 | 91.7 | 90.0 | 91.3 | 4.83 |
| GROK 4 FAST (REASONING) | 90.8 | 91.7 | 94.2 | 85.6 | 90.6 | 0.18 |
| GROK 4 | 90.8 | 92.5 | 95.0 | 83.1 | 90.4 | 7.56 |
| GPT OSS 120B (HIGH) | 90.0 | 90.0 | 91.7 | 85.6 | 89.3 | 0.21 |
| DEEPSEEK-V3.2 (THINK) | 91.7 | 90.0 | 95.8 | 75.6 | 88.3 | 0.22 |
| GPT-5-MINI (HIGH) | 87.5 | 89.2 | 90.0 | 83.1 | 87.5 | 1.09 |
| GLM 4.5 | 93.3 | 77.5 | 92.5 | 71.3 | 83.7 | 1.71 |
| GPT OSS 20B (HIGH) | 89.2 | 75.0 | 85.0 | 72.5 | 80.4 | 0.22 |
| GEMINI-2.5-PRO | 87.5 | 82.5 | 90.0 | 58.1 | 79.5 | 5.02 |
| GPT-5-NANO (HIGH) | 85.0 | 74.2 | 80.8 | 73.8 | 78.4 | 0.40 |
| GLM 4.5 AIR | 83.3 | 69.2 | 90.0 | 70.6 | 78.3 | 0.90 |
| CLAUDE-SONNET-4.5 (THINK) | 84.2 | 67.5 | 90.8 | 66.9 | 77.3 | 9.09 |
| HUMAN (TOP 1%) | 84.4 | 66.8 | N/A | N/A | N/A | N/A |

**Leaderboard**  Results are published on a public leaderboard at `https://matharena.ai`. The interface is designed for ease of use, allowing users to navigate results, inspect individual model outputs, and verify parsing and grading decisions. This enables users to qualitatively analyze the models' performance and verify the correctness of our parser and grading process.

**Variance estimation**  Due to the small size of most competitions, variance estimation is crucial for robust interpretation. We estimate variance for two key metrics: (1) model rank, important for comparative analysis, and (2) raw scores, which reflect absolute performance. To compute a confidence interval for model ranks, we use a paired permutation test to count the number of models significantly better or worse than a given model $m_i$ at a significance level $\alpha$, yielding a confidence interval for its rank. Details of the test can be found in App. C. To compute a confidence interval for the accuracy, we treat each answer as a Bernoulli trial with parameter $\hat{p}$ and compute variance as $\hat{p}(1 - \hat{p})/N$, where $N$ is the number of questions. $\hat{p}$ is estimated using model accuracy.

## 4   Evaluation

In this section, we present our evaluation of leading LLMs on MATHARENA. We also analyze the results to investigate data contamination, performance trends over time, and confidence intervals. Details on accessing the data and code used in our experiments, along with licensing information, are provided in App. A. To facilitate open research, we release all results and raw model responses on our website `https://matharena.ai`.

**Setup**  We evaluated models on the following competitions from 2025:  AIME [2, 3], HMMT [18], BRUMO [4], CMIMC [5], USAMO [26], IMO [12], and Project Euler [11]. Collectively, these competitions span 162 problems covering algebra, combinatorics, geometry, and number theory. USAMO and IMO are proof-based competitions, while the others require numerical final answers. We evaluated over 50 LLMs across all competitions, incurring approximately USD $2,000$ in API query costs for the experiments discussed in this paper, excluding development expenses.

### 4.1   Numerical Answer Competitions

Our final-answer-based evaluation, excluding Project Euler, includes four competitions comprising 130 problems. We focus on non-deprecated models in this section and present full results in App. B. A model is deprecated once a strictly better version from the same provider is released (e.g., O3-MINI is deprecated upon the release of O4-MINI), after which it is excluded from future evaluations.

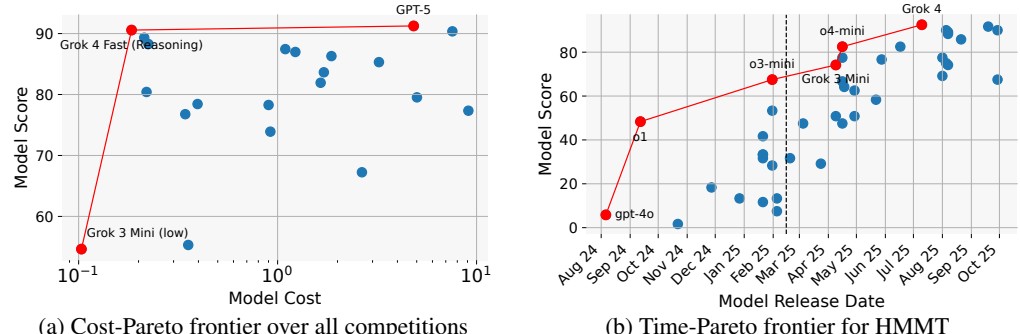

(a) Cost-Pareto frontier over all competitions     (b) Time-Pareto frontier for HMMT

Figure 3: Scores of models with respect to their release date and cost (in USD). Each dot represents a model; the red curves trace the Pareto frontier in both (a) cost vs. score for all competitions, (b) release-date vs. score for HMMT. The black dotted line indicates the release date of HMMT.

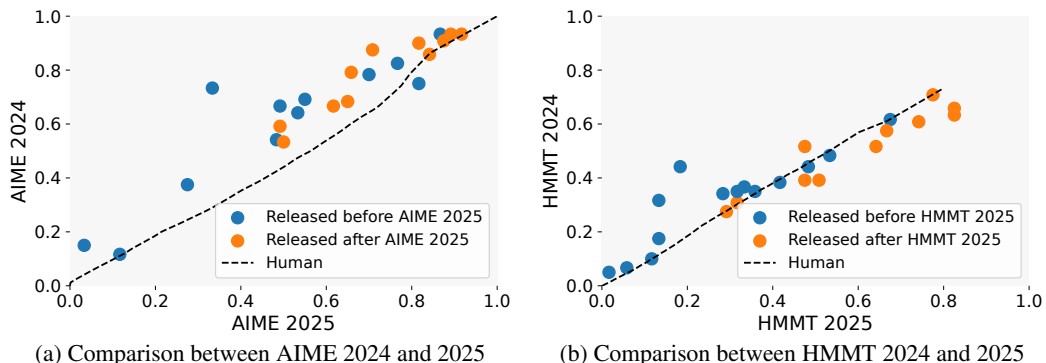

(a) Comparison between AIME 2024 and 2025     (b) Comparison between HMMT 2024 and 2025

Figure 4: Comparison between new and old competitions. The black dotted line indicates quantiles of human performance. Models above the human line are likely contaminated.

**Main results** Table 2 reports results for the best non-deprecated models at the time of writing. Following the evaluation protocol described in Section 3, each model was evaluated four times per problem, with accuracy computed using the pass@1 metric and no additional inference-time strategies (e.g., majority voting). Overall, the latest models demonstrate very strong performance. The best-performing models—GPT-5, GROK 4, and GROK 4 FAST—achieve accuracies of $91.3\%$, $90.6\%$, and $90.4\%$, respectively, with GROK 4 FAST being significantly cheaper. These models vastly outperform the top $1\%$ of human participants in AIME and HMMT, indicating their capability to solve most problems correctly and compete with the best human contestants. Among open-source models, GPT-OSS-120B leads, closely followed by DEEPSEEK-v3.2 (THINK).

**Cost-accuracy Pareto frontier** Fig. 3a shows the cost-accuracy Pareto frontier across all competitions. Cost reflects the money in USD needed to run a model on a full competition, averaged over all competitions. The frontier currently only includes three models from XAI and OPENAI.

**Performance over time** Fig. 3b illustrates the model scores on HMMT 2025 as a function of time. Each dot represents a model release, and the red line denotes the Pareto frontier of accuracy over time. The dashed vertical line marks the competition date, meaning models to the left of it are guaranteed to be uncontaminated. We show similar plots for other competitions in App. B. We observe that models released before September 2024 achieved less than $10\%$ accuracy (e.g., GPT-4o). Performance significantly improved with the release of chain-of-thought reasoning models like O1 and continued to rise with subsequent iterations.

**Data contamination of past competitions** A key aim of our study is to evaluate the reliability of model performance on older competitions, particularly AIME 2024, where contamination may have occurred. Fig. 4a and Fig. 4b compare model scores on the 2024 and 2025 versions of AIME and HMMT. The x-axis shows performance on the 2025 version, while the y-axis shows the 2024

score. The dotted line represents human performance quantiles, enabling us to account for difficulty changes between the years, as the same human quantile is expected to yield similar performance across years. Most models lie above this line on AIME with a margin of $10\% - 20\%$, suggesting inflated performance on AIME 2024 due to data contamination. QwQ-PREVIEW-32B is a notable outlier and outperforms the expected human-aligned performance by nearly $60\%$, indicating extreme contamination. In contrast, the discrepancy is much smaller for HMMT, indicating more trustworthy results—likely because HMMT is less prominent and less likely to be included in training datasets.

Another possible source for contamination of a new competition is that versions of problems from the new competition may have already appeared online, either in past contests or online forums. We investigate this for AIME 2025 and HMMT 2025 using DeepResearch [27], and find that 8 problems from AIME 2025 and 1 problem from HMMT 2025 can be found online in a similar form. We find that these are mostly easier problems that do not affect the overall results, but they underscore an interesting caveat of evaluating in future competitions. Details are provided in App. B.

**Confidence intervals**   Most existing benchmarks for large language models rely on large datasets, raising concerns that the variance in a single competition may be too high to yield meaningful conclusions. In contrast, small competitions are often used to evaluate human participants, indicating that they can be reliable.

Using the methodology from Section 3.4, we compute $95\%$ confidence intervals for rank and accuracy across all competitions. Table 3 shows these intervals averaged across competition, with per-competition intervals shown in App. B. Despite the smaller size, MATHARENA can reliably differentiate between most models. In particular, rank intervals are relatively small, with the top three models being GPT-5, GROK 4 FAST, and GROK 4, all within $1\%$ of each other.

**Repeating runs**   As a more intuitive approach to understanding variance, we follow Abdin et al. [1] and perform repeated evaluations. Specifically, we select several representative models (O4-MINI (MEDIUM), QWEN3-30B-A3B, DEEPSEEK-R1-DISTILL-32B, and DEEPSEEK-R1-DISTILL-14B), sample 100 solutions per problem, and derive 25 score estimates per model using 4 per-problem samples as described in Section 3.4. We then fit kernel density estimates (KDEs) to these score distributions. The results show that the score distributions are sharp, validating our methodology of averaging accuracy over four runs.

Table 3: Variance in the performance of models averaged for all competitions. $95\%$ confidence intervals are shown for both rank and accuracy.

| Model | Rank | Acc (avg) |
|---|---|---|
| GPT-5 (HIGH) | 1-4 | $91.25 \pm 2.4$ |
| GROK 4 FAST (REASONING) | 1-5 | $90.57 \pm 2.5$ |
| GROK 4 | 1-5 | $90.36 \pm 2.5$ |
| GPT OSS 120B (HIGH) | 1-7 | $89.32 \pm 2.6$ |
| DEEPSEEK-V3.2 (THINK) | 2-8 | $88.28 \pm 2.6$ |
| GPT-5-MINI (HIGH) | 4-9 | $87.45 \pm 2.8$ |
| GLM 4.5 | 8-11 | $83.65 \pm 3.0$ |
| GPT OSS 20B (HIGH) | 11-16 | $80.42 \pm 3.4$ |
| GEMINI-2.5-PRO | 11-17 | $79.53 \pm 3.2$ |
| GPT-5-NANO (HIGH) | 12-17 | $78.44 \pm 3.5$ |
| GLM 4.5 AIR | 12-17 | $78.28 \pm 3.5$ |
| CLAUDE-SONNET-4.5 (THINK) | 12-17 | $77.34 \pm 3.5$ |

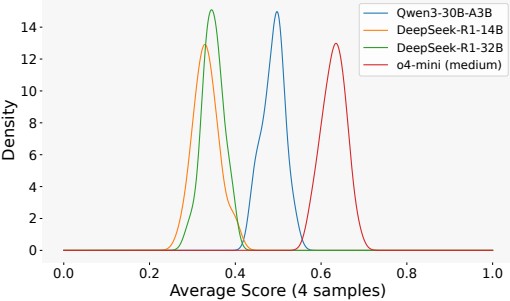

Figure 5: Distribution of the 4-sample accuracy estimates of several models for HMMT.

**Cross-competition correlation**   We additionally compute the Spearman correlation between different competitions. A high correlation indicates consistent model rankings and suggests that a single competition is representative of overall performance. AIME, HMMT, and CMIMC all show correlations above $80\%$, clearly indicating that results from one competition generalize well to other similar competitions. The high overall correlation supports the conclusion that single-competition evaluations are generally robust.

## 4.2   Project Euler

**Setup**   We evaluated six state-of-the-art models on Project Euler: GPT-5, O4-MINI, GROK 4, GROK 4 FAST, GEMINI-2.5-PRO, and CLAUDE-SONNET-4.5. These models were selected based on their

strong performance in other competitions within MATHARENA. Since these problems typically require programming to solve, we allow models to use tools to execute code, as described in Section 3.

**Results**   As shown in Table 4, GPT-5 achieved the highest accuracy of $55\%$, followed by GROK 4 and its faster and cheaper variant at $47.5\%$. CLAUDE-SONNET-4.5 and GEMINI-2.5-PRO lag behind, achieving accuracies of $16.25\%$ and $12.5\%$, respectively.

Table 4: Performance on Project Euler with tools. $95\%$ confidence intervals are shown for both rank and accuracy. Cost reflects the money in USD needed to run a model on all 20 problems.

| Model | Rank | Acc | Cost |
|---|---|---|---|
| GPT-5 (HIGH) | 1-3 | $55.00 \pm 10.9$ | 21.58 |
| GROK 4 FAST | 1-4 | $47.50 \pm 10.9$ | 2.22 |
| GROK 4 | 1-4 | $47.50 \pm 10.9$ | 46.92 |
| O4-MINI (HIGH) | 2-4 | $43.75 \pm 10.9$ | 11.38 |
| CLAUDE-SONNET-4.5 | 5-6 | $16.25 \pm 8.1$ | 16.93 |
| GEMINI-2.5-PRO | 5-6 | $12.50 \pm 7.2$ | 6.90 |

### 4.3 Evaluating Natural Language Proofs

One of the core goals of MATHARENA is to evaluate models on proof-based math competitions, particularly the USAMO [26], IMO [12], and Putnam [25]. Of these, USAMO 2025 and IMO 2025 have occurred at the time of writing. We conducted evaluations immediately after problem release using the procedure described in Section 3. More details about the evaluation for USAMO 2025 can be found in our previous report [28]. In this section, we discuss the results of IMO 2025.

**Model selection and evaluation**   We evaluated six state-of-the-art models: GPT-5, O3, O4-MINI, GEMINI-2.5-PRO, GROK 4, and DEEPSEEK-R1-0528. We applied the best-of-n selection strategy introduced by Dekoninck et al. [9], selecting the best proof from 32 samples per problem. In this process, the model itself serves as a judge in a bracket tournament between the generated proofs, choosing the winner of each round until a final proof is selected. Prompts for this procedure are provided in App. D.

**Results**   GPT-5 achieved the highest score, with an average of $38\%$ (16 points). Although this result may appear modest, especially given the 200 dollars spent to generate only 24 answers, it nonetheless represents strong performance given the exceptional difficulty of the IMO. However, 16 points fall short of the 19 required for a bronze medal (19/42). Full results are available on our leaderboard, where individual responses and judge feedback can be explored in detail. Several examples of model responses are given in App. E. Because the number of problems is small, the rank confidence intervals are wider than in numerical competitions. We therefore recommend caution when interpreting the results, particularly when comparing models with similar scores.

**Qualitative analysis**   We highlight several qualitative findings from our evaluation. First, GROK 4 performed considerably below expectations. Many of its initial responses were extremely brief, often providing only a final answer without explanation. Similar patterns can be seen on other MATHARENA benchmarks, where GROK 4 frequently produces answers with little depth or justification. In contrast, GEMINI-2.5-PRO shows a different issue: when it fails to find a valid proof, it often cites non-existent theorems. This is especially problematic because it misleads users by presenting false authority, thereby undermining trust in the model's reasoning. While this behavior was less common in the IMO responses compared to the USAMO [28], it remains a concern. On a more positive note, compared to earlier evaluations [28], we observed fewer formatting errors and fewer cases of models over-optimizing for final-answer styles, such as boxing entire proofs or assuming that every response must be numerical. This suggests progress in handling open-ended mathematical reasoning tasks more reliably. Finally, one of our judges briefly reviewed a subset of the 32 raw responses produced by the models before the best-of-n selection. They noted that many of these raw responses were very weak and estimated that, without filtering, model scores would likely have dropped below $10\%$. Interestingly, the judge also observed that some unselected answers appeared more coherent than the chosen ones, yet contained more factual errors.

## 5   Discussion

We briefly describe the limitations and broader impact of our work.

**Limitations**   There are only a limited number of annual competitions that are sufficiently challenging to serve as effective benchmarks for state-of-the-art LLMs. As a result, the size of MATHARENA

Table 5: Main results of our evaluation. Problems are scored out of 7 points, with the maximum possible total score being 42. Listed scores are averaged over all four runs. We measure cost in USD, and report the average score across all generations and graders for each problem.

| Model | P1 (/7) | P2 (/7) | P3 (/7) | P4 (/7) | P5 (/7) | P6 (/7) | Total (/42) | Cost (avg) |
|---|---|---|---|---|---|---|---|---|
| GPT-5 (HIGH) | 2.3 | 0.0 | 1.8 | 5.3 | 6.8 | 0.0 | 16.0 | 53.61 |
| GEMINI-2.5-PRO | 1.0 | 0.0 | 5.0 | 3.3 | 4.0 | 0.0 | 13.3 | 107.99 |
| O3 (HIGH) | 0.0 | 0.0 | 0.5 | 2.5 | 4.0 | 0.0 | 7.0 | 55.83 |
| O4-MINI (HIGH) | 1.1 | 0.0 | 0.4 | 3.3 | 1.3 | 0.0 | 6.0 | 25.84 |
| GROK 4 | 0.9 | 0.3 | 1.3 | 0.9 | 1.8 | 0.0 | 5.0 | 131.96 |
| DEEPSEEK-R1-0528 | 0.3 | 0.0 | 0.4 | 0.0 | 2.3 | 0.0 | 2.9 | 14.88 |

remains small, leading to relatively wide confidence intervals in our results. However, we expect this to improve over time as more competitions are added, gradually reducing uncertainty. Furthermore, current state-of-the-art models already solve nearly all but the most difficult questions in answer-based competitions. This suggests that such benchmarks may soon become saturated, possibly as early as 2026. To maintain meaningful evaluations, we anticipate the need to identify or design more challenging competitions. Unlike static benchmarks, however, the dynamic nature of MATHARENA allows it to evolve alongside model capabilities, ensuring continued relevance as the field progresses.

Further, there are some potential concerns about residual data contamination arising from the time gap between a model's release and the competition date. On our leaderboard, we clearly indicate models that were released after the competition data. However, since there is a time gap between the public release of a competition and our evaluation, it is theoretically possible that closed-source models could be updated with the new competition data before we evaluate them. In practice, however, our evaluations are conducted only a few hours to at most a few days after the competition concludes, while current training pipelines require much longer to incorporate new data. For these reasons, we believe that contamination risks in our setting are minimal.

**Broader impact** MATHARENA has already made a notable impact on the field. Several major model providers have cited MATHARENA results in their release notes, including PHI-4-REASONING [1], GEMINI-2.5-PRO [31], and GROK-3 [34]. In February, we were the first to demonstrate that the performance of reasoning-focused LLMs on older math competitions generalizes well to newer ones. Our work has gotten significant community interest, and we expect MATHARENA to remain a valuable and adaptive resource, supporting the ongoing evaluation of LLMs by keeping the benchmark both challenging and aligned with the evolving frontier of model capabilities.

## 6    Conclusion

We introduced MATHARENA, a benchmark designed to evaluate the mathematical performance of large language models (LLMs) using uncontaminated problems from human math competitions. The key insight was that such competitions generate a diverse set of challenging and naturally uncontaminated problems, making them ideal for rigorous evaluation. To support this, we developed a scalable pipeline that parses problems and answers, samples model solutions, extracts final answers, and verifies correctness. Using this framework, we evaluated over 50 LLMs on 162 problems from seven math competitions held in 2025. Our results show substantial progress in LLMs' mathematical capabilities while also confirming the impact of data contamination in prior benchmarks.

## Acknowledgments

This research was partially funded by the Ministry of Education and Science of Bulgaria (support for INSAIT, part of the Bulgarian National Roadmap for Research Infrastructure). This project was supported with computational resources provided by Google Cloud Platform (GCP). This work has received funding from the Swiss State Secretariat for Education, Research and Innovation (SERI) (SERI-funded ERC Consolidator Grant).

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

## A  Code and Data Availability and Reprodicibility

This section outlines the availability of code and data used in our benchmark. Our code is publicly available at `https://github.com/eth-sri/matharena`. Regarding data availability, we typically publish datasets on HuggingFace at `https://huggingface.co/MathArena`. All data is available under the CC-BY-NC-SA 4.0 license, which allows for non-commercial use and modification, provided that the original source is credited. This license was chosen after consultation with the competition organizers. In particular, we reached out to all competition organizers to ensure that our data release complies with their policies. All organizers agreed to the use of their questions under the CC-BY-NC-SA 4.0 license.

## B  Additional Results

**Full main results**    In Table 6 we include the complete results of our benchmark on many proprietary and open-source LLMs. We did not evaluate poorly performing models, or models that were superseded by better versions on the CMIMC or BRUMO competitions.

**Domain-specific results**    While the best-performing models tend to show consistent results across different competitions, their performance varies significantly across mathematical problem domains. We manually classified each problem into one of four standard high-school competition categories: Algebra, Combinatorics, Geometry, and Number Theory, as shown in Table 7. Calculus problems were grouped under Algebra, while non-standard or word-based problems were categorized under Combinatorics.

Table 7: Distribution of problem types per competition. Some problems were assigned multiple domains, as they combined concepts from more than one area.

| Competition | Algebra | Comb. | Geo. | NT |
|---|---|---|---|---|
| AIME | 9 | 9 | 8 | 6 |
| HMMT Feb. | 7 | 10 | 11 | 4 |
| CMIMC | 8 | 14 | 14 | 5 |
| BRUMO | 7 | 10 | 8 | 5 |
| Total | 31 | 43 | 41 | 20 |

As seen in Table 8, nearly all models struggle more with combinatorial and geometric problems—domains that typically require greater creativity capabilities. In the domain of Geometry, models consistently struggle with visualizing constructions or applying synthetic reasoning. Instead, most correct solutions rely on analytical approaches—typically brute-force coordinate methods. As a result, even weaker models can solve simpler problems using these methods. However, when compared to other mathematical domains, stronger models show relatively poorer performance on Geometry tasks, likely because the domain requires spatial intuition and reasoning that current models lack. In contrast, when problems require standard techniques or symbolic manipulation, as is often the case in Algebra and Number Theory, LLMs show significantly stronger performance.

**Confidence intervals per competition**    In Table 9,Table 10, Table 11, and Table 12 we show the confidence intervals for the results of each competition using the method described in Section 3.

**Timeline for all competitions**    In Fig. 6 we show the Pareto frontiers for all competitions in function of time. The red curves trace the Pareto-optimal points in release-date vs. accuracy for all competitions. The black dotted lines mark the competition release dates.

**Token usage per model**    As shown in Table 13, we also tracked the number of tokens used by almost all model during evaluation. This includes both prompt and response tokens, averaged over all problems from final-answer competitions. The higher number of input tokens for GPT-5 is due to caching of response tokens in some cases.

**Data contamination of past competitions**    We used DeepResearch [27] to search the internet for problems similar to those in the AIME 2025 and HMMT 2025 competitions. We found the following sources that may be similar to the problems in the AIME 2025 and HMMT 2025 competitions:

- AIME 2025 Problem 1: `https://www.quora.com/In-what-bases-b-does-b-7-divide-into-9b-7-without-any-remainder`

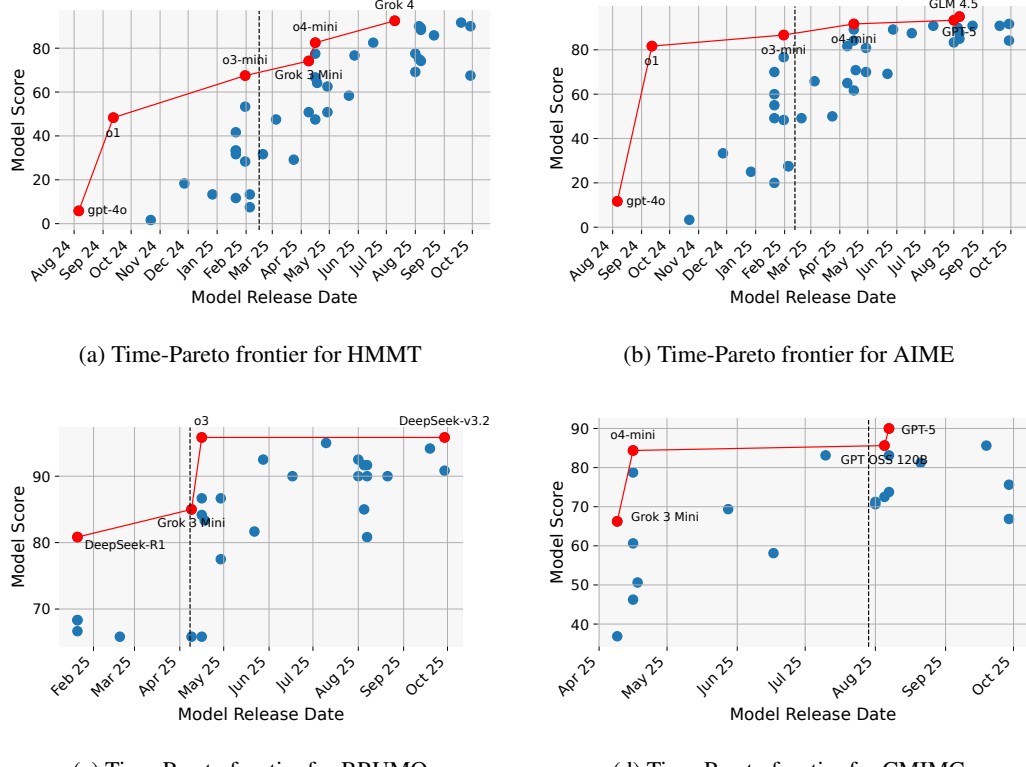

(a) Time-Pareto frontier for HMMT

(b) Time-Pareto frontier for AIME

(c) Time-Pareto frontier for BRUMO

(d) Time-Pareto frontier for CMIMC

Figure 6: Accuracy of models with respect to their release date and cost. Each dot represents a released model; the red curves trace the Pareto-optimal points in release-date vs. accuracy for all competitions. Black dotted lines mark the competition release dates.

- AIME 2025 Problem 3: `https://math.stackexchange.com/questions/3548821/ number-of-combination-of-incremental-numbers-for-a-number` (reported by Dimitris Papailiopoulos)

- AIME 2025 Problem 5: `https://math.stackexchange.com/questions/3146556/ how-many-five-digit-numbers-formed-from-digits-1-2-3-4-5-used-exactly-once-a` (reported by Dimitris Papailiopoulos)

- AIME 2025 Problem 6: `https://www.wyzant.com/resources/answers/105893/ isosceles_trapezoid_abcd_contains_inscribed_circle_o_the_area_of_the_ trapezoid_is_510_square_centimeters_the_radius_of_the_circle_is_10_cm_find_ ad`

- AIME 2025 Problem 10: `https://math.stackexchange.com/questions/1923331/ how-many-different-ways-can-the-numbers-1-9-be-arranged-in-a-3x9-grid`

- AIME 2025 Problem 17: `https://math.stackexchange.com/questions/2526124/ find-all-positive-integers-n-such-that-n-3-divides-n2-27#:~:text=4e`

- AIME 2025 Problem 22: `https://math.stackexchange.com/questions/2145033/ counting-all-sets-with-the-same-least-common-multiple`

- AIME 2025 Problem 25: `https://math.stackexchange.com/questions/4824833/ re-visit-combinatorics-involving-3-adjacent-objects`

- HMMT 2025 Problem 15: `https://puzzling.stackexchange.com/questions/116775/ the-longest-path-of-edges-on-a-3x3-grid`

# C  Permutation Test for Rank Confidence Interval

**Construction confidence interval**    To compute a confidence interval at significance level $\alpha$ for a model's rank, we use pairwise comparisons between models via a paired permutation test.

For a given model $m_i$, we compare it to every other model $m_j$:

- Let $N_{i,j}^+$ be the number of models $m_j$ for which $m_i$ performs significantly better.

- Let $N_{i,j}^-$ be the number of models $m_j$ for which $m_i$ performs significantly worse.

Given a total of $N$ models, the rank confidence interval for model $m_i$ is $[N_{i,j}^+ + 1, N - N_{i,j}^-]$.

**Paired permutation test**    A paired permutation test is a non-parametric method for testing whether two related samples have significantly different means. It requires:

- A set of paired observations: $(x_1, y_1), (x_2, y_2), \ldots, (x_n, y_n)$
- A test statistic $T$ computed over these pairs.

The null hypothesis is that $x_i$ and $y_i$ are exchangeable, i.e., swapping them does not affect the test statistic in expectation.

To test this hypothesis, we generate random permutations by flipping each pair $(x_i, y_i)$ with a probability of 50%. For each permutation, we compute the test statistic. By repeating this process many times, we create a distribution of the test statistic under the null hypothesis. We then compare the unpermuted test statistic to this distribution to determine if it is significantly different. Specifically, if the quantile of the unpermuted statistic within this distribution is less than the significance level $\alpha$, we reject the null hypothesis and conclude that there is a significant difference between the samples.

**Paired permutation test for rank**    In our setting, each paired sample indicates the correctness of a single answer of models $m_i$ and $m_j$ on the same problem. For a single competition, the test statistic is the total difference in performance between the two models:

$$T((x_1, y_1), \ldots, (x_n, y_n)) = \sum_{i=1}^{n}(x_i - y_i).$$

When aggregating performance across multiple competitions, we weight each competition equally, regardless of the number of problems it contains. This results in a weighted sum in the test statistic, where each sample is weighted by the inverse of the number of problems in its respective competition. This ensures that the total weight associated with each competition is equal. If we denote the number of problems in competition $c$ as $N_c$, and $c_i$ as the competition of problem $i$, the test statistic becomes:

$$T((x_1, y_1), \ldots, (x_n, y_n)) = \sum_{i=1}^{n} \frac{(x_i - y_i)}{N_{c_i}}.$$

We can then apply the procedure described above to compute the rank confidence interval for model $m_i$.

Table 6: The full main results of our numerical answer evaluation, sorted by average score. We measure cost in USD, and report the average score across all generations for the 3 competitions.

| Model | AIME | HMMT | BRUMO | CMIMC | Acc (avg) | Cost (avg) |
|---|---|---|---|---|---|---|
| GPT-5 (HIGH) | 95.0 | 88.3 | 91.7 | 90.0 | 91.3 | 4.83 |
| GROK 4 FAST (REASONING) | 90.8 | 91.7 | 94.2 | 85.6 | 90.6 | 0.18 |
| GROK 4 | 90.8 | 92.5 | 95.0 | 83.1 | 90.4 | 7.56 |
| GPT OSS 120B (HIGH) | 90.0 | 90.0 | 91.7 | 85.6 | 89.3 | 0.21 |
| DEEPSEEK-V3.2 (THINK) | 91.7 | 90.0 | 95.8 | 75.6 | 88.3 | 0.22 |
| GPT-5-MINI (HIGH) | 87.5 | 89.2 | 90.0 | 83.1 | 87.5 | 1.09 |
| DEEPSEEK-V3.1 (THINK) | 90.8 | 85.8 | 90.0 | 81.3 | 87.0 | 1.23 |
| O4-MINI (HIGH) | 91.7 | 82.5 | 86.7 | 84.4 | 86.3 | 1.86 |
| O3 (HIGH) | 89.2 | 77.5 | 95.8 | 78.8 | 85.3 | 3.23 |
| GEMINI-2.5-PRO-05-06 | 83.3 | 80.8 | 89.2 | N/A | 84.4 | 0.68 |
| GLM 4.5 | 93.3 | 77.5 | 92.5 | 71.3 | 83.7 | 1.71 |
| DEEPSEEK-R1-0528 | 89.2 | 76.7 | 92.5 | 69.4 | 81.9 | 1.65 |
| GPT OSS 20B (HIGH) | 89.2 | 75.0 | 85.0 | 72.5 | 80.4 | 0.22 |
| GEMINI-2.5-PRO | 87.5 | 82.5 | 90.0 | 58.1 | 79.5 | 5.02 |
| GPT-5-NANO (HIGH) | 85.0 | 74.2 | 80.8 | 73.8 | 78.4 | 0.40 |
| GLM 4.5 AIR | 83.3 | 69.2 | 90.0 | 70.6 | 78.3 | 0.90 |
| CLAUDE-SONNET-4.5 (THINK) | 84.2 | 67.5 | 90.8 | 66.9 | 77.3 | 9.09 |
| O3-MINI (HIGH) | 86.7 | 67.5 | N/A | N/A | 77.1 | 1.92 |
| GROK 3 MINI (HIGH) | 81.7 | 74.2 | 85.0 | 66.3 | 76.8 | 0.34 |
| QWEN3-235B-A22B | 80.8 | 62.5 | 86.7 | N/A | 76.7 | 0.25 |
| K2-THINK | 83.3 | 65.0 | 83.3 | 65.6 | 74.3 | N/A |
| O4-MINI (MEDIUM) | 84.2 | 66.7 | 84.2 | 60.6 | 73.9 | 0.92 |
| QWEN3-A22B-2507-THINK | 92.5 | 71.7 | 45.8 | N/A | 70.0 | 1.34 |
| CLAUDE-OPUS-4.0 (THINK) | 69.2 | 58.3 | 81.7 | N/A | 69.7 | 34.26 |
| GEMINI-2.5-FLASH (THINK) | 70.8 | 64.2 | 83.3 | 50.6 | 67.2 | 2.65 |
| QWEN3-30B-A3B | 70.0 | 50.8 | 77.5 | N/A | 66.1 | 0.15 |
| O3-MINI (MEDIUM) | 76.7 | 53.3 | N/A | N/A | 65.0 | 0.92 |
| O1 (MEDIUM) | 81.7 | 48.3 | N/A | N/A | 65.0 | 24.06 |
| DEEPSEEK-R1 | 70.0 | 41.7 | 80.8 | N/A | 64.2 | 0.72 |
| PHI-4-REASONING-PLUS | 74.2 | 46.7 | N/A | N/A | 60.4 | 0.19 |
| QWQ-32B | 65.8 | 47.5 | N/A | N/A | 56.7 | 0.59 |
| O4-MINI (LOW) | 61.7 | 47.5 | 65.8 | 46.3 | 55.3 | 0.36 |
| GROK 3 MINI (LOW) | 65.0 | 50.8 | 65.8 | 36.9 | 54.6 | 0.10 |
| DEEPSEEK-R1-DISTILL-32B | 60.0 | 33.3 | 68.3 | N/A | 53.9 | 0.16 |
| DEEPSEEK-R1-DISTILL-70B | 55.0 | 33.3 | 66.7 | N/A | 51.7 | 0.19 |
| DEEPSEEK-R1-DISTILL-14B | 49.2 | 31.7 | 68.3 | N/A | 49.7 | 0.08 |
| CLAUDE-3.7-SONNET (THINK) | 49.2 | 31.7 | 65.8 | N/A | 48.9 | 10.89 |
| OPENTHINKER-32B | 56.7 | 36.7 | N/A | N/A | 46.7 | N/A |
| GEMINI-2.0-FLASH-THINKING | 53.3 | 35.8 | N/A | N/A | 44.6 | N/A |
| S1.1-32B | 50.0 | 37.5 | N/A | N/A | 43.8 | N/A |
| DEEPSEEK-V3-03-24 | 50.0 | 29.2 | N/A | N/A | 39.6 | 0.14 |
| LIMO | 49.2 | 30.0 | N/A | N/A | 39.6 | N/A |
| O3-MINI (LOW) | 48.3 | 28.3 | N/A | N/A | 38.3 | 0.34 |
| QWQ-32B-PREVIEW | 33.3 | 18.3 | N/A | N/A | 25.8 | 0.32 |
| GEMINI-2.0-FLASH | 27.5 | 13.3 | N/A | N/A | 20.4 | 0.04 |
| DEEPSEEK-V3 | 25.0 | 13.3 | N/A | N/A | 19.2 | 0.10 |
| GEMINI-2.0-PRO | 27.5 | 7.5 | N/A | N/A | 17.5 | 0.40 |
| DEEPSEEK-R1-DISTILL-1.5B | 20.0 | 11.7 | N/A | N/A | 15.8 | 0.11 |
| LLAMA-4-MAVERICK | 22.5 | 8.3 | N/A | N/A | 15.4 | 0.03 |
| GPT-4O | 11.7 | 5.8 | N/A | N/A | 8.8 | 0.26 |
| CLAUDE-3.5-SONNET | 3.3 | 1.7 | N/A | N/A | 2.5 | 0.26 |

Table 8: Average accuracy per model per problem type, sorted by average score.

| Model | Algebra | Combinatorics | Geometry | Number Theory |
|---|---|---|---|---|
| GPT-5 (HIGH) | 100.0 | 91.3 | 81.1 | 94.0 |
| GROK 4 | 96.8 | 87.8 | 86.6 | 88.1 |
| GPT OSS 120B (HIGH) | 100.0 | 86.0 | 81.1 | 90.5 |
| GPT-5-MINI (HIGH) | 93.5 | 83.7 | 80.5 | 91.7 |
| O4-MINI (HIGH) | 96.0 | 86.6 | 75.6 | 86.9 |
| O3 (HIGH) | 91.1 | 84.9 | 80.5 | 78.6 |
| GEMINI-2.5-PRO-05-06 | 97.8 | 77.6 | 75.0 | 91.7 |
| GLM 4.5 | 94.4 | 77.3 | 75.6 | 85.7 |
| DEEPSEEK-R1-0528 | 89.5 | 80.8 | 73.8 | 79.8 |
| GPT OSS 20B (HIGH) | 93.5 | 75.0 | 71.3 | 82.1 |
| GEMINI-2.5-PRO | 96.8 | 70.3 | 73.2 | 76.2 |
| GLM 4.5 AIR | 87.1 | 69.8 | 75.0 | 84.5 |
| GPT-5-NANO (HIGH) | 86.3 | 73.3 | 70.1 | 88.1 |
| O3-MINI (HIGH) | 84.4 | 72.4 | 71.1 | 80.0 |
| GROK 3 MINI (HIGH) | 83.9 | 76.2 | 67.1 | 78.6 |
| QWEN3-235B-A22B | 90.2 | 67.2 | 68.5 | 86.7 |
| O4-MINI (MEDIUM) | 82.3 | 68.6 | 66.5 | 77.4 |
| CLAUDE-OPUS-4.0 (THINK) | 75.0 | 65.5 | 63.0 | 80.0 |
| QWEN3-A22B-2507-THINK | 87.0 | 57.8 | 59.3 | 78.3 |
| GEMINI-2.5-FLASH (THINK) | 83.1 | 56.4 | 58.5 | 73.8 |
| QWEN3-30B-A3B | 75.0 | 50.9 | 65.7 | 81.7 |
| O3-MINI (MEDIUM) | 70.3 | 56.6 | 64.5 | 70.0 |
| O1 (MEDIUM) | 67.2 | 61.8 | 59.2 | 75.0 |
| DEEPSEEK-R1 | 68.5 | 52.6 | 63.9 | 78.3 |
| PHI-4-REASONING-PLUS | 65.6 | 50.0 | 59.2 | 72.5 |
| QWQ-32B | 59.4 | 47.4 | 55.3 | 72.5 |
| O4-MINI (LOW) | 61.3 | 52.9 | 45.1 | 70.2 |
| GROK 3 MINI (LOW) | 62.9 | 44.2 | 45.7 | 70.2 |
| DEEPSEEK-R1-DISTILL-32B | 56.5 | 42.2 | 54.6 | 71.7 |
| DEEPSEEK-R1-DISTILL-70B | 51.1 | 37.1 | 54.6 | 76.7 |
| DEEPSEEK-R1-DISTILL-14B | 50.0 | 37.1 | 50.9 | 73.3 |
| CLAUDE-3.7-SONNET (THINK) | 53.3 | 41.4 | 43.5 | 61.7 |
| OPENTHINKER-32B | 54.7 | 26.3 | 53.9 | 65.0 |
| GEMINI-2.0-FLASH-THINKING | 65.6 | 21.1 | 40.8 | 70.0 |
| S1.1-32B | 56.2 | 19.7 | 51.3 | 60.0 |
| DEEPSEEK-V3-03-24 | 40.6 | 19.7 | 50.0 | 55.0 |
| LIMO | 43.8 | 23.7 | 42.1 | 62.5 |
| O3-MINI (LOW) | 39.1 | 23.7 | 46.1 | 52.5 |
| QWQ-32B-PREVIEW | 29.7 | 7.9 | 31.6 | 40.0 |
| GEMINI-2.0-FLASH | 20.3 | 5.3 | 26.3 | 30.0 |
| DEEPSEEK-V3 | 17.2 | 3.9 | 28.9 | 30.0 |
| GEMINI-2.0-PRO | 21.9 | 3.9 | 18.4 | 35.0 |
| DEEPSEEK-R1-DISTILL-1.5B | 9.4 | 5.3 | 18.4 | 35.0 |
| LLAMA-4-MAVERICK | 14.1 | 2.6 | 17.1 | 40.0 |
| GPT-4O | 4.7 | 0.0 | 10.5 | 27.5 |
| CLAUDE-3.5-SONNET | 1.6 | 0.0 | 3.9 | 5.0 |
| OVERALL | 64.5 | 48.8 | 55.5 | 68.6 |

Table 9: Results of the BRUMO competition with 95% confidence intervals.

| Model | Rank | Acc (avg) |
|---|---|---|
| O3 (HIGH) | 1-10 | $95.83 \pm 3.6$ |
| DEEPSEEK-V3.2 (THINK) | 1-9 | $95.83 \pm 3.6$ |
| GROK 4 | 1-12 | $95.00 \pm 3.9$ |
| GROK 4 FAST | 1-14 | $94.17 \pm 4.2$ |
| DEEPSEEK-R1-0528 | 1-14 | $92.50 \pm 4.7$ |
| GLM 4.5 | 1-14 | $92.50 \pm 4.7$ |
| GPT-5 (HIGH) | 1-17 | $91.67 \pm 4.9$ |
| GPT OSS 120B (HIGH) | 1-17 | $91.67 \pm 4.9$ |
| CLAUDE-SONNET-4.5 (THINK) | 1-17 | $90.83 \pm 5.2$ |
| GPT-5-MINI (HIGH) | 3-21 | $90.00 \pm 5.4$ |
| DEEPSEEK-V3.1 (THINK) | 4-20 | $90.00 \pm 5.4$ |
| GEMINI-2.5-PRO | 3-20 | $90.00 \pm 5.4$ |
| GLM 4.5 AIR | 2-19 | $90.00 \pm 5.4$ |
| GEMINI-2.5-PRO-05-06 | 4-21 | $89.17 \pm 5.6$ |
| QWEN3-235B-A22B | 7-23 | $86.67 \pm 6.1$ |
| O4-MINI (HIGH) | 7-24 | $86.67 \pm 6.1$ |
| GROK 3 MINI (HIGH) | 10-24 | $85.00 \pm 6.4$ |
| GPT OSS 20B (HIGH) | 7-25 | $85.00 \pm 6.4$ |
| O4-MINI (MEDIUM) | 9-25 | $84.17 \pm 6.5$ |
| GEMINI-2.5-FLASH (THINK) | 12-25 | $83.33 \pm 6.7$ |
| K2-THINK | 11-25 | $83.33 \pm 6.7$ |
| CLAUDE-OPUS-4.0 (THINK) | 15-25 | $81.67 \pm 6.9$ |
| DEEPSEEK-R1 | 16-25 | $80.83 \pm 7.0$ |
| GPT-5-NANO (HIGH) | 15-25 | $80.83 \pm 7.0$ |
| QWEN3-30B-A3B | 18-25 | $77.50 \pm 7.5$ |
| DEEPSEEK-R1-DISTILL-14B | 26-31 | $68.33 \pm 8.3$ |
| DEEPSEEK-R1-DISTILL-32B | 26-31 | $68.33 \pm 8.3$ |
| DEEPSEEK-R1-DISTILL-70B | 26-31 | $66.67 \pm 8.4$ |
| GROK 3 MINI (LOW) | 26-31 | $65.83 \pm 8.5$ |
| O4-MINI (LOW) | 26-31 | $65.83 \pm 8.5$ |
| CLAUDE-3.7-SONNET (THINK) | 26-31 | $65.83 \pm 8.5$ |
| QWEN3-A22B-2507-THINK | 32-32 | $45.83 \pm 8.9$ |

Table 10: Results of the CMIMC competition with 95% confidence intervals.

| Model | Rank | Acc (avg) |
|---|---|---|
| GPT-5 (HIGH) | 1-4 | $90.00 \pm 4.6$ |
| GROK 4 FAST | 1-8 | $85.62 \pm 5.4$ |
| GPT OSS 120B (HIGH) | 1-7 | $85.62 \pm 5.4$ |
| O4-MINI (HIGH) | 1-8 | $84.38 \pm 5.6$ |
| GROK 4 | 2-8 | $83.12 \pm 5.8$ |
| GPT-5-MINI (HIGH) | 2-8 | $83.12 \pm 5.8$ |
| DEEPSEEK-V3.1 (THINK) | 2-9 | $81.25 \pm 6.0$ |
| O3 (HIGH) | 3-12 | $78.75 \pm 6.3$ |
| DEEPSEEK-V3.2 (THINK) | 7-14 | $75.62 \pm 6.7$ |
| GPT-5-NANO (HIGH) | 8-15 | $73.75 \pm 6.8$ |
| GPT OSS 20B (HIGH) | 8-17 | $72.50 \pm 6.9$ |
| GLM 4.5 | 8-17 | $71.25 \pm 7.0$ |
| GLM 4.5 AIR | 9-17 | $70.62 \pm 7.1$ |
| DEEPSEEK-R1-0528 | 9-17 | $69.38 \pm 7.1$ |
| CLAUDE-SONNET-4.5 (THINK) | 10-18 | $66.88 \pm 7.3$ |
| GROK 3 MINI (HIGH) | 11-18 | $66.25 \pm 7.3$ |
| K2-THINK | 11-18 | $65.62 \pm 7.4$ |
| O4-MINI (MEDIUM) | 15-19 | $60.62 \pm 7.6$ |
| GEMINI-2.5-PRO | 18-19 | $58.13 \pm 7.6$ |
| GEMINI-2.5-FLASH (THINK) | 20-21 | $50.62 \pm 7.7$ |
| O4-MINI (LOW) | 20-21 | $46.25 \pm 7.7$ |
| GROK 3 MINI (LOW) | 22-22 | $36.88 \pm 7.5$ |

Table 11: Results of the AIME competition with 95% confidence intervals.

| Model | Rank | Acc (avg) |
|---|---|---|
| GPT-5 (HIGH) | 1-9 | 95.00 ± 3.9 |
| GLM 4.5 | 1-12 | 93.33 ± 4.5 |
| QWEN3-A22B-2507-THINK | 1-12 | 92.50 ± 4.7 |
| O4-MINI (HIGH) | 1-15 | 91.67 ± 4.9 |
| DEEPSEEK-V3.2 (THINK) | 1-15 | 91.67 ± 4.9 |
| DEEPSEEK-V3.1 (THINK) | 1-16 | 90.83 ± 5.2 |
| GROK 4 | 1-16 | 90.83 ± 5.2 |
| GROK 4 FAST | 1-18 | 90.83 ± 5.2 |
| GPT OSS 120B (HIGH) | 1-18 | 90.00 ± 5.4 |
| DEEPSEEK-R1-0528 | 2-18 | 89.17 ± 5.6 |
| GPT OSS 20B (HIGH) | 2-18 | 89.17 ± 5.6 |
| O3 (HIGH) | 2-18 | 89.17 ± 5.6 |
| GPT-5-MINI (HIGH) | 4-22 | 87.50 ± 5.9 |
| GEMINI-2.5-PRO | 4-23 | 87.50 ± 5.9 |
| O3-MINI (HIGH) | 4-24 | 86.67 ± 6.1 |
| GPT-5-NANO (HIGH) | 6-24 | 85.00 ± 6.4 |
| O4-MINI (MEDIUM) | 8-24 | 84.17 ± 6.5 |
| CLAUDE-SONNET-4.5 (THINK) | 8-25 | 84.17 ± 6.5 |
| K2-THINK | 13-25 | 83.33 ± 6.7 |
| GLM 4.5 AIR | 13-25 | 83.33 ± 6.7 |
| GEMINI-2.5-PRO-05-06 | 13-25 | 83.33 ± 6.7 |
| O1 (MEDIUM) | 13-25 | 81.67 ± 6.9 |
| GROK 3 MINI (HIGH) | 14-25 | 81.67 ± 6.9 |
| QWEN3-235B-A22B | 14-25 | 80.83 ± 7.0 |
| O3-MINI (MEDIUM) | 18-30 | 76.67 ± 7.6 |
| PHI-4-REASONING-PLUS | 25-30 | 74.17 ± 7.8 |
| GEMINI-2.5-FLASH (THINK) | 25-32 | 70.83 ± 8.1 |
| QWEN3-30B-A3B | 25-33 | 70.00 ± 8.2 |
| DEEPSEEK-R1 | 25-33 | 70.00 ± 8.2 |
| CLAUDE-OPUS-4.0 (THINK) | 25-34 | 69.17 ± 8.3 |
| QWQ-32B | 27-34 | 65.83 ± 8.5 |
| GROK 3 MINI (LOW) | 27-35 | 65.00 ± 8.5 |
| O4-MINI (LOW) | 28-37 | 61.67 ± 8.7 |
| DEEPSEEK-R1-DISTILL-32B | 30-37 | 60.00 ± 8.8 |
| OPENTHINKER-32B | 33-41 | 56.67 ± 8.9 |
| DEEPSEEK-R1-DISTILL-70B | 33-43 | 55.00 ± 8.9 |
| GEMINI-2.0-FLASH-THINKING | 33-43 | 53.33 ± 8.9 |
| DEEPSEEK-V3-03-24 | 35-43 | 50.00 ± 8.9 |
| S1.1-32B | 35-43 | 50.00 ± 8.9 |
| LIMO | 36-43 | 49.17 ± 8.9 |
| DEEPSEEK-R1-DISTILL-14B | 36-43 | 49.17 ± 8.9 |
| CLAUDE-3.7-SONNET (THINK) | 35-43 | 49.17 ± 8.9 |
| O3-MINI (LOW) | 35-43 | 48.33 ± 8.9 |
| QWQ-32B-PREVIEW | 44-45 | 33.33 ± 8.4 |
| GEMINI-2.0-PRO | 44-48 | 27.50 ± 8.0 |
| GEMINI-2.0-FLASH | 45-48 | 27.50 ± 8.0 |
| DEEPSEEK-V3 | 45-49 | 25.00 ± 7.7 |
| LLAMA-4-MAVERICK | 45-49 | 22.50 ± 7.5 |
| DEEPSEEK-R1-DISTILL-1.5B | 47-49 | 20.00 ± 7.2 |
| GPT-4O | 50-50 | 11.67 ± 5.7 |
| CLAUDE-3.5-SONNET | 51-51 | 3.33 ± 3.2 |

Table 12: Results of the HMMT competition with 95% confidence intervals.

| Model | Rank | Acc (avg) |
|---|---|---|
| GROK 4 | 1-6 | 92.50 ± 4.7 |
| GROK 4 FAST (REASONING) | 1-7 | 91.67 ± 4.9 |
| GPT OSS 120B (HIGH) | 1-7 | 90.00 ± 5.4 |
| DEEPSEEK-V3.2 (THINK) | 1-7 | 90.00 ± 5.4 |
| GPT-5-MINI (HIGH) | 1-8 | 89.17 ± 5.6 |
| GPT-5 (HIGH) | 1-10 | 88.33 ± 5.7 |
| DEEPSEEK-V3.1 (THINK) | 2-10 | 85.83 ± 6.2 |
| O4-MINI (HIGH) | 6-13 | 82.50 ± 6.8 |
| GEMINI-2.5-PRO | 5-14 | 82.50 ± 6.8 |
| GEMINI-2.5-PRO-05-06 | 7-16 | 80.83 ± 7.0 |
| GLM 4.5 | 8-17 | 77.50 ± 7.5 |
| O3 (HIGH) | 8-17 | 77.50 ± 7.5 |
| DEEPSEEK-R1-0528 | 8-17 | 76.67 ± 7.6 |
| GPT OSS 20B (HIGH) | 10-21 | 75.00 ± 7.7 |
| GROK 3 MINI (HIGH) | 10-21 | 74.17 ± 7.8 |
| GPT-5-NANO (HIGH) | 9-21 | 74.17 ± 7.8 |
| QWEN3-A22B-2507-THINK | 11-23 | 71.67 ± 8.1 |
| GLM 4.5 AIR | 12-24 | 69.17 ± 8.3 |
| O3-MINI (HIGH) | 14-25 | 67.50 ± 8.4 |
| CLAUDE-SONNET-4.5 (THINK) | 14-24 | 67.50 ± 8.4 |
| O4-MINI (MEDIUM) | 15-25 | 66.67 ± 8.4 |
| K2-THINK | 17-25 | 65.00 ± 8.5 |
| GEMINI-2.5-FLASH (THINK) | 17-25 | 64.17 ± 8.6 |
| QWEN3-235B-A22B | 18-26 | 62.50 ± 8.7 |
| CLAUDE-OPUS-4.0 (THINK) | 21-28 | 58.33 ± 8.8 |
| O3-MINI (MEDIUM) | 24-32 | 53.33 ± 8.9 |
| QWEN3-30B-A3B | 25-32 | 50.83 ± 8.9 |
| GROK 3 MINI (LOW) | 25-32 | 50.83 ± 8.9 |
| O1 (MEDIUM) | 26-33 | 48.33 ± 8.9 |
| QWQ-32B | 26-33 | 47.50 ± 8.9 |
| O4-MINI (LOW) | 26-33 | 47.50 ± 8.9 |
| PHI-4-REASONING-PLUS | 26-33 | 46.67 ± 8.9 |
| DEEPSEEK-R1 | 29-36 | 41.67 ± 8.8 |
| S1.1-32B | 33-39 | 37.50 ± 8.7 |
| OPENTHINKER-32B | 33-42 | 36.67 ± 8.6 |
| GEMINI-2.0-FLASH-THINKING | 33-43 | 35.83 ± 8.6 |
| DEEPSEEK-R1-DISTILL-70B | 34-43 | 33.33 ± 8.4 |
| DEEPSEEK-R1-DISTILL-32B | 34-43 | 33.33 ± 8.4 |
| DEEPSEEK-R1-DISTILL-14B | 35-43 | 31.67 ± 8.3 |
| CLAUDE-3.7-SONNET (THINK) | 34-43 | 31.67 ± 8.3 |
| LIMO | 35-43 | 30.00 ± 8.2 |
| DEEPSEEK-V3-03-24 | 35-43 | 29.17 ± 8.1 |
| O3-MINI (LOW) | 36-43 | 28.33 ± 8.1 |
| QWQ-32B-PREVIEW | 44-46 | 18.33 ± 6.9 |
| GEMINI-2.0-FLASH | 44-48 | 13.33 ± 6.1 |
| DEEPSEEK-V3 | 44-48 | 13.33 ± 6.1 |
| DEEPSEEK-R1-DISTILL-1.5B | 45-49 | 11.67 ± 5.7 |
| LLAMA-4-MAVERICK | 45-50 | 8.33 ± 4.9 |
| GEMINI-2.0-PRO | 47-50 | 7.50 ± 4.7 |
| GPT-4O | 48-51 | 5.83 ± 4.2 |
| CLAUDE-3.5-SONNET | 50-51 | 1.67 ± 2.3 |

Table 13: Average number of input and output tokens used per model across all final-answer competitions.

|  | Input | Output |
|---|---|---|
| GPT OSS 20B (HIGH) | 151 | 44 214 |
| GPT-5-NANO (HIGH) | 186 | 31 651 |
| GLM 4.5 AIR | 156 | 25 488 |
| GLM 4.5 | 156 | 24 174 |
| CLAUDE-3.7-SONNET (THINK) | 199 | 24 168 |
| DEEPSEEK-R1-0528 | 141 | 23 855 |
| GPT OSS 120B (HIGH) | 151 | 23 597 |
| GEMINI-2.5-FLASH (THINK) | 144 | 22 803 |
| GROK 3 MINI (HIGH) | 141 | 21 917 |
| DEEPSEEK-R1-DISTILL-1.5B | 174 | 20 106 |
| CLAUDE-SONNET-4.5 (THINK) | 203 | 19 157 |
| QWEN3-A22B-2507-THINK | 158 | 18 749 |
| DEEPSEEK-V3.1 (THINK) | 152 | 18 562 |
| PHI-4-REASONING-PLUS | 393 | 18 309 |
| DEEPSEEK-V3.2 (THINK) | 152 | 17 426 |
| QWEN3-30B-A3B | 158 | 17 024 |
| GPT-5-MINI (HIGH) | 197 | 16 904 |
| GROK 4 | 151 | 16 697 |
| QWQ-32B | 191 | 16 142 |
| GEMINI-2.5-PRO | 156 | 16 133 |
| GPT-5 (HIGH) | 1782 | 15 390 |
| CLAUDE-OPUS-4.0 (THINK) | 199 | 15 189 |
| O3-MINI (HIGH) | 170 | 14 526 |
| QWEN3-235B-A22B | 158 | 14 063 |
| O1 (MEDIUM) | 172 | 13 324 |
| GROK 4 FAST (REASONING) | 264 | 12 277 |
| DEEPSEEK-R1 | 145 | 11 038 |
| DEEPSEEK-R1-DISTILL-70B | 148 | 10 566 |
| QWQ-32B-PREVIEW | 191 | 8808 |
| O3-MINI (MEDIUM) | 172 | 6892 |
| O4-MINI (MEDIUM) | 189 | 6549 |
| GROK 3 MINI (LOW) | 140 | 6369 |
| DEEPSEEK-V3-03-24 | 167 | 3828 |
| GEMINI-2.5-PRO-05-06 | 151 | 3712 |
| GEMINI-2.0-FLASH | 172 | 3040 |
| GEMINI-2.0-PRO | 173 | 2644 |
| O3-MINI (LOW) | 172 | 2511 |
| DEEPSEEK-V3 | 167 | 2462 |
| O4-MINI (LOW) | 148 | 2433 |
| LLAMA-4-MAVERICK | 170 | 1252 |
| GPT-4O | 173 | 814 |
| CLAUDE-3.5-SONNET | 193 | 542 |
| GEMINI-2.0-FLASH-THINKING | 350 | 498 |

## D Prompts

We provide the full set of prompts used in our evaluation below. In each prompt, {{problem}} is replaced with the problem statement.

---

**Prompt AIME 2025**

```
Please reason step by step, and put your final answer within \boxed{}.The answer is an integer between 0
and 999 inclusive.

{{problem}}
```

---

**Prompt CMIMC and HMMT and BRUMO 2025**

```
Please reason step by step, and put your final answer within \boxed{}.

{{problem}}
```

---

**Prompt Project Euler**

```
You are solving a problem from Project Euler.
As a tool you can use, you are given access to a code execution environment. Invoke that tool to execute
Python or C++ code. You can execute code up to 20 times, so you can use them quite liberally.
You can also use the tool to run code that helps you reason about the problem, but does not directly
compute the final answer.
Answers that consist of code are not accepted, I will only accept a final answer to the question that does
not require me to run any code you produce, as you should use the tool for this.
You are REQUIRED to use the tool to compute the final answer. It is impossible to solve this problem
without using the tool.
Instead, your code should compute the answer (via the tool), after which you should put your final answer
within \\boxed{{}}.
Your answer should be correctly formatted by putting your final answer within \boxed{}, i.e., end your
reply with "### Final answer: \\boxed{your_answer_here}".

{{problem}}
```

---

**Tool Description Project Euler**

```
function:
    description: 'Executes the code in the given language and returns the standard
        output and standard error. Your code is always executed as a self-contained
        script, and it does not have access to the previously executed code blocks!
        If you use python, your code will be run in an environment with the following
        libraries installed: pandas, numpy, scikit-learn, sympy, gmpy2'
    name: execute_code
    parameters:
    properties:
        code:
            description: 'The self-contained code to execute'
            type: string
        lang:
            description: 'The programming language of the code (python or cpp)'
            type: string
    required:
    -code
    -lang
    type: object
    type: function
```

---

**Prompt USAMO 2025**

```
Give a thorough answer to the following question. Your answer will be graded by human judges based on
accuracy, correctness, and your ability to prove the result. You should include all steps of the proof. Do
not skip important steps, as this will reduce your grade. It does not suffice to merely state the result.
Use LaTeX to format your answer.

{{problem}}
```

## Prompt IMO 2025

Your task is to write a proof solution to the following problem. Your proof will be graded by human judges for accuracy, thoroughness, and clarity. When you write your proof, follow these guidelines:

- You are creating a proof, not a proof outline. Each step should be carefully explained and documented. If not properly explained, the judge will assume that you cannot explain it, and therefore decrease your grade.
- You can use general theorems and lemmas, but only if they are well-known. As a rule of thumb: if the result has a name and is famous enough to have a Wikipedia page or something similar to describe it, it is allowed. Any result from papers that would not be taught in high-school or low-level bachelor courses in mathematics should not be used. Any use of such results will immediately give you a zero grade.
- Do not skip computation steps in your proof. Clearly explain what transformations were done and why they are allowed in each step of a calculation.
- You should use correct LaTeX notation to write equations and mathematical symbols. You should encompass these equations in appropriate symbols ("\\(" and "\\)" for inline math, "\\[" and "\\]" for block math) to enhance the clarity of your proof. Do not use any unicode characters.
- Your proof should be self-contained.
- If you are not sure about a specific step, or do not know how to prove an intermediate result, clearly state this. It is much preferable to indicate your uncertainty rather than making incorrect statements or claims.

{{problem}}

## Prompt Best-of-n Selection IMO 2025

You are judging which of the two LLM-generated proofs for a given math problem is better.

### Input:

Your input will consist of the following components:
- **Problem Statement**: A mathematical problem that the proof is attempting to solve.
- **Proof Solution A/B**: The proofs that you need to evaluate. This proof may contain errors, omissions, or unclear steps. Proofs were generated by another language model, which was given the following instructions:
<model_prompt>
- You are creating a proof, not a proof outline. Each step should be carefully explained and documented. If not properly explained, the judge will assume that you cannot explain it, and therefore decrease your grade.
- You can use general theorems and lemmas, but only if they are well-known. As a rule of thumb: if the result has a name and is famous enough to have a Wikipedia page or something similar to describe it, it is allowed. Any result from papers that would not be taught in high-school or low-level bachelor courses in mathematics should not be used. Any use of such results will immediately give you a zero grade.
- Do not skip computation steps in your proof. Clearly explain what transformations were done and why they are allowed in each step of a calculation.
- You should use correct LaTeX notation to write equations and mathematical symbols. You should encompass these equations in appropriate symbols ("\\(" and "\\)" for inline math, "\\[" and "\\]" for block math) to enhance the clarity of your proof. Do not use any unicode characters.
- Your proof should be self-contained.
- If you are not sure about a specific step, or do not know how to prove an intermediate result, clearly state this. It is much preferable to indicate your uncertainty rather than making incorrect statements or claims.
</model_prompt>

### How the solution should be graded:
The following examples are small mistakes that should only be slightly penalised:
- Makes a small computational mistake that can be easily fixed
- Misses an edge case which can be easily proven/disproven
- Skips over a step that follows without much reasoning or manual work
On the other hand, a solution should should be severely penalised if:
- It marks a step as trivial, if it is not immediately obvious with little reasoning why this would be the case.
- It omits algebra-heavy computational steps, regardless of whether or not it has outlined the methodology. Skipping shorter computations should be permitted.
- Generalizes over a pattern without rigorously describing the pattern, or without proving any relevant properties.
- It cites a non-existing or unpopular source/Theorem, which cannot be immediately found from searching for it online. Thus, any theorems that can be immediately found and have a Wikipedia article are allowed.

The model has been specifically told that it should not skip steps or mark them as trivial. Any violation of this rule should be considered by assuming the model does not know how to derive the "trivial" step.

### Further Potential Issues:

Here are some common types of issues to look for:

- **Overgeneralization**: The generated proof proceeds by proving the problem in one or more specific cases, and then concludes that the result holds in general. However, it does not provide a proof for the general case.
- **Oversimplification**: The proof marks steps as trivial or obvious without proper justification.
- **Skipping Computation Steps**: Proofs that skip computation steps or do not explain transformations clearly can lead to misunderstandings.
- **Citing Non-Standard Works or Theorems**: Some models may cite theorems or results that are not well-known or are not typically taught in high-school or low-level bachelor courses. Such theorems are only allowed if they are well known.
- **Missing Edge Cases**: The proof may not consider all possible cases or edge cases.

### Scoring instructions

You should compare the two proofs and determine which one is better. If you believe Proof A is better, end your analysis with \boxed{A}. If you believe Proof B is better, end your analysis with \boxed{B}. If you believe both proofs are equally good, end your analysis with \boxed{equal}.

# E    Example Problems and Model Solutions

## E.1    AIME 2025 - Problem 15

**Problem**

Let $N$ denote the numbers of ordered triples of positive integers $(a, b, c)$ such that $a, b, c \leq 3^6$ and $a^3 + b^3 + c^3$ is a multiple of $3^7$. Find the remainder when $N$ is divided by 1000.

**Solution by GPT-5**

Let $3^7 = 2187$. For any $1 \leq n \leq 3^6$, write uniquely $n = u + 27v$ with $0 \leq u, v \leq 26$ (the pair $(u, v) = (0, 0)$ corresponds to $n = 3^6$). Then

$$n^3 \equiv u^3 + 81u^2v \pmod{2187}.$$

Thus for a triple $(x, y, z)$ with $(u_i, v_i)$ the corresponding pairs, the condition

$$x^3 + y^3 + z^3 \equiv 0 \pmod{2187}$$

becomes

$$A + 81B \equiv 0 \pmod{2187}, \quad \text{where } A = \sum u_i^3 \text{ and } B = \sum u_i^2 v_i.$$

Fix $u = (u_1, u_2, u_3)$ and write $d = \gcd(27, u_1^2, u_2^2, u_3^2)$. Then the linear congruence $\sum u_i^2 v_i \equiv -A/81 \pmod{27}$ has:

- $27^2$ solutions if $d = 1$, provided $81 \mid A$;
- $27^2 \cdot 9$ solutions if $d = 9$, provided $729 \mid A$;
- $27^3$ solutions if $d = 27$, provided $2187 \mid A$.

So it remains to count the number of $u$-triples in each case. Write $u_i$ in base 3 as

$$u_i = a_i + 3b_i + 9c_i \quad \text{with } a_i, b_i, c_i \in \{0, 1, 2\}.$$

Then:

- $u_i$ divisible by 3 iff $a_i = 0$; divisible by 9 iff $a_i = b_i = 0$.
- $u_i^3 \equiv a_i^3 + 9a_i^2b_i + 27(\dots) \pmod{81}$;
- If $a_i = 0$, then $u_i^3 \equiv 27b_i^3 + 243b_i^2c_i \pmod{729}$;
- If $a_i = b_i = 0$, then $u_i^3 \equiv 729c_i^3 \pmod{2187}$.

Count by cases:

1. $d = 1$ (at least one $u_i \not\equiv 0 \pmod 3$), requiring $81 \mid A$. Modulo 9, $a_i^3 \in \{0, 1, 8\}$, and $\sum a_i^3 \equiv 0 \pmod 9$ with some $a_i \neq 0$ forces $\{a_1, a_2, a_3\} = \{0, 1, 2\}$. For such a pattern, the further mod 81 condition reduces to a single congruence modulo 9 in the $(b_i, c_i)$. For $a_i = 1$ or 2, the contribution is uniformly distributed over $\mathbb{Z}/9$ as $(b_i, c_i)$ vary; for $a_i = 0$ it is in $\{0, 3, 6\}$ equally often. Hence for each of the $3! = 6$ assignments of $(0, 1, 2)$ to the three places, exactly 81 of the $9^3$ choices of $(b_i, c_i)$ work. Thus $N_1 = 6 \cdot 81 = 486$.

2. $d = 9$ (all $u_i \equiv 0 \pmod 3$ but not all $\equiv 0 \pmod 9$), requiring $729 \mid A$. With $a_i = 0$, $u_i^3 \equiv 27b_i^3 + 243b_i^2 c_i \pmod{729}$, so

$$729 \mid A \iff \sum b_i^3 + 9 \sum b_i^2 c_i \equiv 0 \pmod{27}.$$

   Among the $9^3$ choices of $(b_i, c_i)$, exactly 81 satisfy this (by a simple check or character sum), of which 27 have all $b_i = 0$ (these correspond to $d = 27$). Therefore $N_9 = 81 - 27 = 54$.

3. $d = 27$ (all $u_i \equiv 0 \pmod 9$), requiring $2187 \mid A$. Then $u_i = 9c_i$ and $A = 729 \sum c_i^3$; so

$$2187 \mid A \iff \sum c_i \equiv 0 \pmod 3.$$

   There are $N_{27} = 3^2 = 9$ such triples.

Therefore the total number of $(x, y, z)$ is

$$N = 27^2 N_1 + 27^2 \cdot 9 N_9 + 27^3 N_{27} = 729 \cdot 486 + 729 \cdot 9 \cdot 54 + 27^3 \cdot 9 = 885735.$$

Hence $N \pmod{1000}$ is 735.

$$\boxed{735}$$

## E.2   HMMT Feb 2025 - Problem 20

**Problem**

The circumference of a circle is divided into 45 arcs, each of length 1. Initially, there are 15 snakes, each of length 1, occupying every third arc. Every second, each snake independently moves either one arc left or one arc right, each with probability $\frac{1}{2}$. If two snakes ever touch, they merge to form a single snake occupying the arcs of both of the previous snakes, and the merged snake moves as one snake. Compute the expected number of seconds until there is only one snake left.

**Solution by GROK 4 FAST**

$$\boxed{\dfrac{448}{3}}$$

**Explanation**

The problem involves 15 snakes on a circle of 45 arcs, initially placed every third arc, resulting in 15 gaps of 2 empty arcs each. Define $h_i = G_i/2$ for each gap $G_i$, so initially all $h_i = 1$ and $\sum h_i = 15$. The process tracks the number of clusters $k$ (equal to the number of gaps) until $k = 1$.

At each step, $k$ independent directions $d_1, \ldots, d_k \in \{+1, -1\}$ are chosen with equal probability. The updates are

$$\varepsilon_i = \frac{d_{i+1} - d_i}{2}, \quad i = 1, \ldots, k \text{ (cyclic)},$$

so

$$h_i' = h_i + \varepsilon_i.$$

A gap is removed if $h_i' \leq 0$, which occurs only if $h_i = 1$ and $\varepsilon_i = -1$, reducing $k$ by the number of such events. The sum of the remaining $h_i$ remains 15.

Thus the state evolves as a Markov chain on partitions of 15, where the number of parts is $k$.

Computations for small $n$ (number of snakes) yield expected times:

$$e_2 = 2, \quad e_3 = \tfrac{16}{3}, \quad e_4 = 10.$$

These fit the quadratic form

$$e_n = \frac{2}{3}(n^2 - 1),$$

verified as

$$e_2 = \tfrac{2}{3}(4 - 1) = 2, \quad e_3 = \tfrac{2}{3}(9 - 1) = \tfrac{16}{3}, \quad e_4 = \tfrac{2}{3}(16 - 1) = 10.$$

For $n = 15$,

$$e_{15} = \frac{2}{3}(225 - 1) = \frac{2}{3} \times 224 = \frac{448}{3}.$$

This pattern arises from the coalescing dynamics, where the expected time scales quadratically with $n$ due to the symmetric random walk behavior of the gaps.

## E.3 IMO 2025 - Problem 1

### Problem

A line in the plane is called
emphsunny if it is not parallel to any of the $x$-axis, the $y$-axis, or the line $x + y = 0$.
Let $n \geq 3$ be a given integer. Determine all nonnegative integers $k$ such that there exist $n$ distinct lines in the plane satisfying both of the following:

- for all positive integers $a$ and $b$ with $a + b \leq n + 1$, the point $(a, b)$ lies on at least one of the lines; and

- exactly $k$ of the $n$ lines are sunny.

### Solution by O4-MINI

**Proof.** Fix an integer $n \geq 3$. Let

$$S_n = \{(a, b) \in \mathbb{Z}^2 : a \geq 1, \, b \geq 1, \, a + b \leq n + 1\},$$

so $|S_n| = \frac{n(n+1)}{2}$. We seek $n$ distinct lines in the plane, exactly $k$ of which are "sunny," that together contain every point of $S_n$. Recall a line is "sunny" if it is not parallel to any of

- the $x$–axis,

- the $y$–axis,

- the line $x + y = 0$.

Equivalently, "non–sunny" lines are exactly the horizontals, verticals, or antidiagonals $x + y = $ const.
Write $s = n - k$ for the number of non–sunny lines, and $k$ for the sunny lines. We will show:

1. One can cover $S_n$ with $n$ lines having exactly $k \in \{0, 1, 3\}$ sunny lines.

2. No other value of $k$ is possible.

—

**Step 1 - Reduction to a smaller triangular grid.** Let $\mathcal{L}$ be any collection of $n$ lines covering $S_n$, with $s$ of them non–sunny and $k = n - s$ sunny. Since non–sunny lines are parallel to one of the three directions (horizontal, vertical, or antidiagonal $x + y = $ const), each non–sunny line covers points in at most one of the three "grid directions." One checks easily that to maximize the number of points of $S_n$ covered by $s$ non–sunny lines, one should take them all horizontal:

$$y = 1, \, y = 2, \, \ldots, \, y = s.$$

Indeed any other choice covers no more than this. These $s$ horizontal lines cover exactly

$$\sum_{i=1}^{s} \left| \{(a, i) : a \geq 1, \, a + i \leq n + 1\} \right| = \sum_{i=1}^{s} (n + 1 - i) = s(n + 1) - \tfrac{s(s+1)}{2}.$$

Since $|S_n| = \frac{n(n+1)}{2}$, the number of points not yet covered is
$$\frac{n(n+1)}{2} - \left[ s(n+1) - \frac{s(s+1)}{2} \right] = \frac{(n-s)(n-s+1)}{2} = \frac{k(k+1)}{2}.$$
One checks that the uncovered points are precisely
$$T_k \;=\; \{(a,b) \in \mathbb{Z}^2 : \; a \geq 1, \; b \geq s+1, \; a+b \leq n+1\}.$$
If we set $b' = b - s$, then $b' = 1, 2, \ldots, k$ and
$$a + b \leq n+1 \iff a + b' \leq k+1,$$
so the uncovered set is congruent to the standard "triangular grid"
$$S_k = \{(a,b') \in \mathbb{Z}^2 : \; a \geq 1, \; b' \geq 1, \; a+b' \leq k+1\},$$
which has exactly $k(k+1)/2$ points. To finish covering $S_n$ we must cover $T_k$, i.e. cover $S_k$, by the remaining $k$ lines—and these $k$ must be sunny, because we have already used up all $s$ non–sunny lines.

Thus the original problem for $(n, k)$ is solvable if and only if the smaller problem

> "Cover the triangular grid $S_k$ by $k$ lines, none parallel to the three forbidden directions"

admits a solution.

——

**Step 2 - Characterization of which $k$ work for $S_k$.** Write
$$S_k = \{(a,b) \in \mathbb{Z}^2 : \; a \geq 1, \; b \geq 1, \; a+b \leq k+1\},$$
and let
$$B = \{(a,b) \in S_k : \; a = 1 \; \vee \; b = 1 \; \vee \; a+b = k+1\}$$
be the boundary lattice points of the little triangle. One checks
$$|B| = (k) + (k) + (k) - 3 = 3k - 3$$
because the three corners $(1,1), (1,k), (k,1)$ are each counted twice.
Now observe:

- A line that is not parallel to any of the three boundary–directions (horizontal, vertical, or antidiagonal) meets the perimeter of the convex triangle in exactly two points (entry and exit). Hence it can pass through at most two boundary lattice points of $S_k$.

Therefore $k$ such "sunny" lines can together cover at most $2k$ of the $3k - 3$ boundary points. To cover all of $S_k$ in particular they must cover all boundary points, so one needs
$$2k \;\geq\; 3k - 3 \implies k \leq 3.$$
Hence for any $k \geq 4$, it is impossible to cover $S_k$ by $k$ sunny lines.
It remains only to check $k = 0, 1, 2, 3$:

$k = 0$. Then $S_0$ is empty, and $0$ lines do cover it trivially.

$k = 1$. Then $S_1 = \{(1,1)\}$. A single non–horizontal, non–vertical, non–antidiagonal line through $(1,1)$ covers it.

$k = 2$. Then
$$S_2 = \{(1,1), (1,2), (2,1)\}.$$
But any two of these three points lie on a line of slope $0$, $\infty$, or $-1$, all of which are forbidden. Hence no one sunny line can ever pass through two of them; two distinct sunny lines could cover at most two points of $S_2$, so can never cover all three. Thus $k = 2$ is impossible.

$k = 3$. Here
$$S_3 = \{(1,1), (1,2), (1,3), (2,1), (2,2), (3,1)\}.$$
One checks directly that the three lines
$$L_1 : \; y = -2x + 5 \quad \text{(through } (1,3), (2,1)),$$
$$L_2 : \; y = -\tfrac{1}{2}x + \tfrac{5}{2} \quad \text{(through } (1,2), (3,1)),$$
$$L_3 : \; y = x \quad \text{(through } (1,1), (2,2))$$
have slopes $-2, -\tfrac{1}{2}, 1$, none of which is $0, -1$, or $\infty$, and together they pass through all six points of $S_3$. Hence $k = 3$ is possible.

Combining these cases, one sees that $S_k$ can be covered by $k$ sunny lines if and only if

$$k \in \{0, 1, 3\}.$$

—

**Step 3 - Lifting back to the original $n$.** From Step 1 the existence of an $n$–line cover of $S_n$ with exactly $k$ sunny lines is equivalent to the existence of a $k$–line sunny cover of $S_k$. By Step 2 the latter happens precisely for $k = 0, 1, 3$. Finally one checks directly that for every $n \geq 3$ one can realize

$$k = 0 : \quad \text{take horizontals } y = 1, 2, \ldots, n,$$

$$k = 1 : \quad \text{take } y = 1, 2, \ldots, n - 1 \text{ (horizontals) plus any one sunny line through } (1, n),$$

$$k = 3 : \quad \text{take } y = 1, 2, \ldots, n - 3 \quad \text{and then the three sunny lines } \begin{cases} y = -2x + (n + 2), \\ y = -\frac{1}{2} x + \frac{2n-1}{2}, \\ y = x + (n - 3), \end{cases}$$

which cover exactly the top little triangle of size 3 as in Step 2, and hence together cover all of $S_n$.

**Answer.** For each integer $n \geq 3$, there exist $n$ distinct lines covering all points $(a, b)$ with $a, b \geq 1$, $a + b \leq n + 1$ and having exactly $k$ sunny lines if and only if

$$k \in \{0, 1, 3\}.$$

No other nonnegative integer $k$ is possible. This completes the proof. $\qquad\square$

---

**Hypotenuse points in the $n - 1$ boundary lines case**
**Description:** Arguing that if $n - 1$ lines cover the leftmost and bottommost points, then the inside points of the hypotenuse cannot be among the covered ones.
**Points:** 1 / 1
**Comment:** Done for $k$ instead of $n$, but the argument is the needed one.

**Finishing**
**Description:** Finishing by showing that the aforementioned points cannot be covered by one line.
**Points:** 0 / 1
**Comment:** Not present.

**Total:** 2 / 7 points.

## E.4   IMO 2025 - Problem 3

### Problem

Let $\mathbb{N}$ denote the set of positive integers. A function $f : \mathbb{N} \to \mathbb{N}$ is said to be bonza if

$$f(a) \quad \text{divides} \quad b^a - f(b)^{f(a)}$$

for all positive integers $a$ and $b$.
Determine the smallest real constant $c$ such that $f(n) \leqslant cn$ for all bonza functions $f$ and all positive integers $n$.

### Solution by GEMINI-2.5-PRO

An elegant solution to the problem.
We are looking for the smallest real constant $c$ such that $f(n) \leqslant cn$ for all bonza functions $f$ and all positive integers $n$. This constant $c$ is given by

$$c = \sup_{f,n} \frac{f(n)}{n},$$

where the supremum is taken over all bonza functions $f$ and all $n \in \mathbb{N}$.
Let the given condition be

$$P(a, b) : f(a) \mid b^a - f(b)^{f(a)}.$$

#### Part 1: Initial properties of bonza functions

**Lemma 1:** For any bonza function $f$, we have $f(1) = 1$.
*Proof:* Using $P(1, 1)$, we have

$$f(1) \mid 1^1 - f(1)^{f(1)},$$

which simplifies to $f(1) \mid 1 - f(1)^{f(1)}$. Since $f(1) \geq 1$, we know $f(1)^{f(1)}$ is divisible by $f(1)$. As $f(1)$ divides both $f(1)^{f(1)}$ and $1 - f(1)^{f(1)}$, it must divide their sum, which is 1. Thus $f(1) = 1$.

**Lemma 2:** For any bonza function $f$ and any $a \in \mathbb{N}$, we have $f(a) \mid a^a$.
*Proof:* Using $P(a, a)$, we have $f(a) \mid a^a - f(a)^{f(a)}$. Since $f(a) \geq 1$, $f(a)^{f(a)}$ is divisible by $f(a)$. As $f(a)$ divides both $f(a)^{f(a)}$ and $a^a - f(a)^{f(a)}$, it must divide $a^a$. Thus, $f(a) \mid a^a$.
An important consequence of Lemma 2 is that any prime divisor of $f(a)$ must also be a prime divisor of $a$.

#### Part 2: Analysis based on prime values

Let $p$ be a prime number. By Lemma 2, $f(p) \mid p^p$. This means $f(p)$ must be of the form $p^k$ for some integer $k \geq 0$.

**Lemma 3:** If $p$ is a prime such that $f(p) > 1$, then $f(b) \equiv b \pmod{p}$ for all $b \in \mathbb{N}$.

*Proof:* Suppose $f(p) = p^k$ with $k \geq 1$. The condition $P(p,b)$ states $f(p) \mid b^p - f(b)^{f(p)}$. Thus $b^p - f(b)^{f(p)} \equiv 0 \pmod{p}$. By Fermat's Little Theorem, $b^p \equiv b \pmod{p}$. Hence

$$b - f(b)^{p^k} \equiv 0 \pmod{p}.$$

But $f(b)^{p^k} \equiv f(b) \pmod{p}$ by repeated application. Thus $f(b) \equiv b \pmod{p}$.

Let $S$ be the set of prime numbers $p$ for which $f(p) > 1$. We analyze the structure of $S$.

**Case 1: $S$ contains two or more primes**

Suppose $p, q \in S$ with $p \neq q$. From Lemma 3 applied to $p$, $f(q) \equiv q \pmod{p}$. Since $f(q) = q^j$ for some $j \geq 1$, we have $q^j \equiv q \pmod{p}$, i.e., $p \mid q^{j-1} - 1$. If $S$ is infinite, this forces $j = 1$. Hence $f(q) = q$ for all $q \in S$. Thus $f(p) = p$ for all primes $p$. Then Lemma 3 gives $f(n) \equiv n \pmod{p}$ for all primes $p$. So $f(n) = n$. This $f(n) = n$ is a bonza function. For this function, $f(n)/n = 1$, so $c \geq 1$.

**Case 2: $S$ finite and non-empty**

Let $S = \{p_1, \ldots, p_m\}$ and $P = p_1 p_2 \cdots p_m$. For $q \notin S$, we have $f(q) = 1$. Lemma 3 gives $1 \equiv q \pmod{p_i}$ for each $p_i \in S$. So $q \equiv 1 \pmod{P}$. By Dirichlet's theorem, if $P > 2$, there are infinitely many primes $q \not\equiv 1 \pmod{P}$. Contradiction. Thus $P \leq 2$. So $S$ is either empty or $\{2\}$.

**Case 3: $S$ empty**

If $f(p) = 1$ for all primes $p$, then for any $n > 1$, $f(n)$ must divide $p^n - 1$ for all primes $p$. If a prime $q$ divides $f(n)$, take $p = q$. Then $q \mid q^n - 1$, impossible. Thus $f(n) = 1$ for all $n$. This $f(n) = 1$ is a bonza function, giving $f(n)/n = 1/n \leq 1$.

**Part 3: The case $S = \{2\}$ and the value of $c$**

Now let $S = \{2\}$.

- $f(2) > 1$. By Lemma 2, $f(2) \mid 4$, so $f(2) \in \{2, 4\}$.
- $f(p) = 1$ for all odd primes $p$.
- By Lemma 3, $f(b) \equiv b \pmod{2}$. So $f(b)$ is odd for odd $b$, and even for even $b$.

For odd $n$, $f(n) = 1$. For even $n = 2^k m$ with $m$ odd, $f(n)$ must be a power of 2, say $2^s$, and must divide $3^n - 1$. It can be shown that $s \leq v_2(3^n - 1) = k + 2$. So $f(n) \leq 2^{v_2(n)+2}$. Then

$$\frac{f(n)}{n} \leq \frac{2^{k+2}}{2^k m} = \frac{4}{m}.$$

Since $m \geq 1$, this is at most 4. Thus $c \leq 4$.

**Part 4: Construction of a bonza function achieving the bound**

Define

$$f(n) = \begin{cases} 1 & \text{if } n \text{ odd,} \\ 4 & \text{if } v_2(n) = 1, \\ 2^{v_2(n)+2} & \text{if } v_2(n) \geq 2. \end{cases}$$

It can be checked case by case that this is a bonza function. For $n = 4$, $f(4) = 16$, giving $f(4)/4 = 4$.

**Conclusion**

We have shown that for any bonza function $f$ and any $n$, $f(n) \leq 4n$, so $c \leq 4$. We have also constructed an example with $f(4)/4 = 4$, so $c \geq 4$. Therefore the smallest real constant $c$ is

$$\boxed{4}.$$

$f(p) \neq 1$ **is true for finitely many** $p$
**Description:** Showing that there are only a finite number of primes $p$, for which $f(p) \neq 1$.
**Points:** 1 / 1
**Comment:** Correct.

$f(2k + 1) = 1$
**Description:** Showing that if $f$ is not the identity, then $f(a) = 1$ for any odd $a$.
**Points:** 2 / 2
**Comment:** Correct.

$f(2k) = 2^m$
**Description:** Proving that for any even $a$, $f(a)$ is a power of 2.
**Points:** 1 / 1
**Comment:** Correct.

$\nu_2(f(a)) \leq \nu_2(a) + 2$
**Description:** Showing that the power of 2 in $f(a)$ does not exceed the power of 2 of $a$ by more than 2.
**Points:** 1 / 1
**Comment:** Correct.

**Construction for** $c = 4$
**Description:** Deriving a construction for $f$ for which the inequality for $c = 4$ is satisfied tightly.
**Points:** 2 / 2
**Comment:** Correct.

**Total:** 7 / 7 points.

