# OpenReview forum: "MathArena: Evaluating LLMs on Uncontaminated Math Competitions"
_NeurIPS.cc/2025/Datasets_and_Benchmarks_Track — NeurIPS 2025 Datasets and Benchmarks Track poster_

### Official Review · Reviewer_wcTJ · 2025-07-01

**Ethics Flags:** Data privacy, copyright, and consent
**Rating:** 4
**Confidence:** 4

**Summary:**

MathArena introduces a novel benchmark for evaluating LLMs on uncontaminated mathematical reasoning tasks. It leverages newly released math competition problems (e.g., AIME, USAMO, SMT) to ensure models cannot rely on memorization. The benchmark includes 149 problems from five 2025 competitions, evaluating 30 LLMs across both final-answer questions (e.g., algebra, combinatorics) and proof-based problems (e.g., USAMO). The framework includes a scalable pipeline for parsing, evaluating, and ranking models with statistical rigor.

**Dataset Code Accessibility:**

Yes

**Ethical Comments:**

The copyright of the latest exams needs to be considered.

**Ethical Considerations:**

Yes, there are ethics concerns that require attention by the authors

**Limitations Weaknesses:**

1. Scalability of Proof Evaluation: Human grading is labor-intensive and non-scalable. The USAMO 2025 evaluation was limited to 10 models due to expert review costs. No automated proof verification (e.g., via Lean), restricting benchmark growth.
2. Contamination Not Fully Eliminated: Another method to avoid contamination is to build a private leaderboard without answer release. Some studies have already made some attempts, such as OlympicArena[2] (also missed in the reference).


[2] OlympicArena: Benchmarking Multi-discipline Cognitive Reasoning for Superintelligent AI.

**Strengths Contributions:**

1. Contamination-Free Design: uses time-released competitions (problems published after model training) as a natural barrier against data leakage, although there are some existing practices, such as LiveCodeBench[1] (missed in the references) and LiveBench.
2. Inclusion of proof-based problems: include proof-based problems (USAMO 2025), exposing LLMs' severe limitations in rigorous proof generation (<25% accuracy).
3. Rigorous Methodology: Automated parsing for final answers + expert human grading for proofs (reducing false positives). Confidence intervals for ranks/scores, permutation tests, and multi-run sampling address small-sample concerns.

[1] LiveCodeBench: Holistic and Contamination Free Evaluation of Large Language Models for Code.

---

> ### Author Rebuttal · Authors · 2025-07-30
>
> Thank you for your review. We will include LiveCodeBench and OlympicArena in our related work section. Below we provide our responses to the key questions.
>
> **Q1: How about evaluating Lean based solutions?**
>
> Our main reason for not including Lean-based problems is the error-prone, difficult, and time-consuming effort of formalizing the problem statements. For instance, PutnamBench [1], one of the most popular benchmarks for formalized mathematics, reports it took them months of work to formalize 650 statements with a team of seven people, all of whom had experience with mathematics, computer science, and formal proof assistants. Furthermore, current public Lean-based solutions lag significantly behind natural language systems, as reported by [2]. For instance, the team behind Deepseek-Prover V2 [3] formalized 15 selected problems from AIME 24/25 and could solve only 6. This is a rather poor score, especially considering that AIME is the easiest benchmark in our collection. Private Lean-based AI systems from DeepMind and ByteDance can indeed solve challenging olympiad problems, but we currently do not have access to these systems.
>
>
> **Q2: Could you comment on the scalability of the proof-based evaluation?**
>
> Human proof-based evaluation is difficult and takes effort, but we showed in recent work [2] that we could scale up the process to incorporate over 1,000 questions and 5,000 proofs, making us confident we can keep evaluating frontier models on this task. Indeed, we have recently evaluated the most capable models on the IMO and we will incorporate this analysis in the final revision of the paper. More importantly, it is necessary to do this evaluation. Final-answer questions are only a small portion of mathematics, and proof-based evaluation is a necessary development. The initially poor performance on the USAMO, coupled with the qualitative analysis with actionable insights and the subsequent rise in performance reported by several big model providers, indicates that people care about this aspect of mathematics. Finally, we have shown that LLMs are approaching human level capabilities at judging proof correctness [2] and can already assist humans at this task, so we believe that there are avenues in future work through which we could significantly scale up evaluation of proofs.
>
> **Q3: Could you discuss differences with benchmarks that avoid contamination by not releasing answers, e.g. OlympicArena?**
>
> OlympicArena does not release answers, but uses problems from past competitions whose solutions are already readily available elsewhere on the internet and are susceptible for data contamination (as their analysis in Section 4.5 shows). In our introduction, we mention two benchmarks which keep both problems and answers private (FrontierMath and HLE). As mentioned there, these benchmarks are expensive to construct (500 000 USD for HLE) and make it difficult for the broader community to verify the results, thus hurting transparency and reproducibility. In contrast, MathArena engages with the broader community and is fully open, opening the possibility for third-party analysis and verification.
>
> **Q4: Did you consider the licenses associated with the datasets used?**
>
> Yes, as mentioned in Appendix A, we proactively contacted all competition organizers to ask for permission to use their data and abided by the license they required us to use (CC-BY-NC-SA-4.0).
>
> **Additional comments:**
> Finally, we want to highlight our continued effort over the past few months to keep our dynamic benchmark up to date, as promised in the paper. Specifically, we’ve added the IMO 2025 as a new proof-based benchmark, included the latest Project Euler problems to test mathematical coding skills on more complex challenges, and added another final-answer competition, CMIMC 2025.
>
>
> [1] Tsoukalas, George, et al. "Putnambench: Evaluating neural theorem-provers on the putnam mathematical competition." Advances in Neural Information Processing Systems 37 (2024): 11545-11569.
>
> [2] Dekoninck, Jasper, et al. "The Open Proof Corpus: A Large-Scale Study of LLM-Generated Mathematical Proofs." arXiv preprint arXiv:2506.21621 (2025).
>
> [3] Ren, Z. Z., et al. "Deepseek-prover-v2: Advancing formal mathematical reasoning via reinforcement learning for subgoal decomposition." arXiv preprint arXiv:2504.21801 (2025).
>
> [4] ByteDance Seed Team. ByteDance Seed Prover Achieves Silver Medal Score in IMO 2025. Published July 23, 2025.
>
> [5] Google DeepMind Team. AI achieves silver-medal standard solving International Mathematical Olympiad problems. Published July 25, 2024

---

> > ### Comment · Reviewer_wcTJ · 2025-08-05
> >
> > Thank you for your response. I appreciate your efforts in keeping the benchmarks up to date; your contributions are invaluable in measuring the true progress of AI and greatly benefit the community.

---

### Official Review · Reviewer_BSM8 · 2025-07-01

**Rating:** 5
**Confidence:** 3

**Summary:**

This paper presents MathArena, a dynamic benchmark designed to assess the mathematical reasoning abilities of LLMs using uncontaminated, real-time math competition problems. By sourcing questions from newly released contests (e.g., AIME 2025, USAMO 2025), MathArena avoids data leakage and enables forward-looking evaluation. Unlike most existing benchmarks, MathArena includes both final-answer and proof-based tasks, with the latter graded by expert human evaluators. The authors assess 30 LLMs across five competitions, revealing that top models perform strongly on numerical tasks but continue to struggle with proof generation. The benchmark is open-source, reproducible, and continuously updated, providing a transparent alternative to existing static or proprietary benchmarks.

**Dataset Code Accessibility:**

Yes

**Ethical Considerations:**

No, there are no or only very minor ethics concerns

**Final Justification:**

I maintain my current rating.

**Limitations Weaknesses:**

I have a few questions and suggestions regarding this paper:

* How is the cost computed for open-source models? Are these models evaluated via API access or self-deployment? It would also be helpful to report the token usage for different models to provide a clearer picture of computational cost.
* Since the inclusion of proof-based evaluation is a key contribution of this work, I suggest adding concrete examples in the main paper or the appendix. These could include a representative question, a sample model-generated proof, and the corresponding human evaluation.
* Related Work on Contamination: I recommend incorporating relevant prior work on evaluating and mitigating data contamination in math reasoning benchmarks, such as [1][2][3].

[1] Dynamath: A dynamic visual benchmark for evaluating mathematical reasoning robustness of vision language models. ICLR 2025.

[2] Gsm-symbolic: Understanding the limitations of mathematical reasoning in large language models. ICLR 2025.

[3] MATH-Perturb: Benchmarking LLMs' Math Reasoning Abilities against Hard Perturbations[J]. arXiv preprint arXiv:2502.06453, 2025.

**Strengths Contributions:**

* Overall, this paper is well-written and easy to follow.
* MathArena uses newly released math competition problems to ensure uncontaminated assessment of LLMs.
* It is the first benchmark to evaluate LLMs on natural language proofs, capturing deeper mathematical reasoning.
* The benchmark is open-source, reproducible, and offers a comprehensive evaluation of state-of-the-art LLMs.

---

> ### Author Rebuttal · Authors · 2025-07-30
>
> Thank you for your review. Below we provide our responses to the key questions.
>
> **Q1: How is the cost computed for open-source models? Are these models evaluated via API access or self-deployment? Could you provide the token usage for different models to provide a clearer picture of computational cost?**
>
> The token cost for open models is computed using the cheapest provider available for the model on OpenRouter. We evaluated some of the models using self-deployment, but in such cases we use the same cost computation method as if we evaluated them on OpenRouter.
>
> We provide token usage for all models in the table below (each cell shows input tokens / output tokens of the model on a given competition).
>
> | Model | AIME | BRUMO | HMMT | SMT |
> |------|------|------|------|------|
> | Claude-3.7-Sonnet (Think) | 275/24602 | 151/22001 | 169/25902 | 195/22813 |
> | DeepSeek-R1 | 210/11211 | 104/9086 | 121/12817 | 140/10323 |
> | DeepSeek-R1-Distill-14B | 221/12352 | 110/10864 | 127/15559 | 147/12399 |
> | DeepSeek-R1-Distill-32B | 224/12749 | 112/10884 | 129/15555 | 149/11380 |
> | DeepSeek-R1-Distill-70B | 213/10488 | 107/9313 | 124/11898 | 144/10049 |
> | Grok 3 Mini (high) | 211/18460 | 107/14850 | 122/21000 | 141/17682 |
> | Grok 3 Mini (low) | 211/5707 | 107/5423 | 122/6636 | 143/6086 |
> | Qwen3-235B-A22B | 228/14907 | 115/12185 | 132/15098 | 152/13024 |
> | Qwen3-30B-A3B | 228/17970 | 115/14564 | 132/18539 | 152/14870 |
> | gemini-2.5-flash (think) | 223/23871 | 106/21389 | 123/27168 | 142/21599 |
> | gemini-2.5-pro | 223/13397 | 106/17840 | 123/12875 | 143/18603 |
> | o3 (high) | 1225/11885 | 110/10065 | 4069/13786 | 1259/8566 |
> | o4-mini (high) | 695/13982 | 7807/7492 | 511/17637 | 147/10311 |
> | o4-mini (low) | 218/2605 | 110/1874 | 126/2663 | 147/1997 |
> | o4-mini (medium) | 218/6221 | 110/4824 | 126/7298 | 147/4590 |
>
>
>
> **Q2: Could you add representative samples for the proof based questions (problem statement, proof, and human evaluation) to the paper?**
>
> Yes, we will add these in the next revision.
>
> **Q3: Could you incorporate prior work [1, 2, 3] on eliminating contamination in math benchmarks?**
>
> Yes, we will incorporate those works in our related work. Conceptually, we believe those represent an orthogonal approach towards uncontaminated evaluation by focusing on sampling different variants of already existing questions. However, this approach does not fully eliminate contamination. Variants of existing samples almost always need to be solved using the same high-level reasoning approach as the original problem. In contrast, we use completely new problems that require a new high-level reasoning approach. Potential future work might evaluate combinations of new problems and their perturbations.
>
> **Additional comments:**
> Finally, we want to highlight our continued effort over the past few months to keep our dynamic benchmark up to date, as promised in the paper. Specifically, we’ve added the IMO 2025 as a new proof-based benchmark, included the latest Project Euler problems to test mathematical coding skills on more complex challenges, and added another final-answer competition, CMIMC 2025.

---

> > ### Comment · Reviewer_BSM8 · 2025-08-04
> >
> > Thank you for your detailed rebuttal. Including a discussion of prior work on mitigating data contamination, as well as using representative samples for proof-based questions, would further strengthen the paper. I will maintain my current positive rating and encourage the authors to incorporate these additions in the revision.

---

### Official Review · Reviewer_smDA · 2025-07-03

**Rating:** 4
**Confidence:** 4

**Summary:**

The authors present MATHARENA, a benchmark for rigorously evaluating LLMs on mathematical reasoning using uncontaminated problems from human math competitions, which provide diverse and challenging test cases. They develop a scalable pipeline to parse problems, sample and evaluate model solutions, and verify correctness systematically. Applying this framework, they evaluate 30 LLMs on 149 problems from five 2025 math competitions, revealing notable improvements in LLM mathematical capabilities while also confirming that data contamination has inflated results in prior benchmarks. Additionally, the dataset is fully open-sourced, benefiting the reproducibility of the results.

**Additional Feedback:**

Please see the weakness and strength part.

**Dataset Code Accessibility:**

Yes

**Dataset Code Comments:**

The authors provide a fully open-source dataset, along with a well-documented README and a result visualization website.

**Ethical Considerations:**

No, there are no or only very minor ethics concerns

**Final Justification:**

Thank you to the authors for their responses.

Regarding the potential contamination issue, I suggest incorporating the relevant discussion into the revised manuscript. For the failure case analysis, it would also be beneficial to include a more detailed discussion in the main content, as this can help the research community further improve the math and general reasoning performance of LLMs.

After reading the authors' responses and the comments from other reviewers, I will keep my score as 4.

**Limitations Weaknesses:**

(1) Timing Sensitivity: The uncontaminated nature depends on evaluation timing versus model training schedules, which may erode if models rapidly incorporate new data. I suggest authors discuss more on how to continuously obtain an uncontaminated math reasoning evaluation dataset.

(2) Limited Error Analysis: The paper does not deeply analyze why models fail on certain problems, missing insights into reasoning bottlenecks that could guide future LLM improvement. A more comprehensive failure mode analysis will be beneficial for understanding.

**Strengths Contributions:**

(1) Contamination-Free Benchmarking: MATHARENA uses newly released math competition problems to ensure evaluations are free from training data leakage.

(2) Rich Problem Diversity: The provided dataset incorporates a wide range of challenging, human-designed problems across multiple competitions to test various mathematical reasoning skills.

(3) Automated Evaluation Pipeline: This paper provides a scalable system for parsing problems, sampling model solutions, and verifying correctness systematically. In addition, the authors make the code fully released, and provide detailed instructions on evaluation as well as a clear visualization of evaluation results.

(4) Comprehensive Evaluated Model Coverage: The authors benchmark 30 different LLMs, offering comprehensive insights into current capabilities across the model landscape.

---

> ### Author Rebuttal · Authors · 2025-07-30
>
> Thank you for your review. Below we provide our responses to the key questions.
>
> **Q1: Could you discuss the potential contamination issues due to the relationship between evaluation and training schedules of LLMs?**
>
> First, we actively contact competition organisers to try to obtain problem statements before they become public. For the SMT and CMIMC (newly added to our leaderboard), we were able to obtain the problem statements this way, preventing any possibility of contamination by model providers. Furthermore, we typically perform our evaluation a few hours to at most a few days after the competition has been held, and current training schedules take a much longer time. Indeed, data needs to be incorporated into the reinforcement learning pipeline and evaluation needs to be performed to make sure performance has not deteriorated on other benchmarks. Even if in the future it becomes possible to update the model this quickly, it is not an easy task to ensure that contamination is not too obvious: if the model’s performance is completely out-of-line with the model’s performance as reported by end users or other benchmarks, suspicions will be raised. Ensuring that the contamination is performed in such a way as to subtly improve performance in a matter of hours seems quite infeasible. Finally, we hope and believe that model providers would not resort to such techniques to improve benchmark performance. The reputational damage can be huge, as evidenced by the Llama-4 problems on LM Arena [1].
>
> **Q2: Could you provide analysis and insights why models fail on certain problems?**
>
> Yes, we will include such an analysis using our recent evaluation of the IMO in the next revision. We provide a short summary of what we plan to include below.
>
> First, for easier competitions, related work [2] has analyzed mistakes of LLMs on AIME I where the failure modes were errors made in enumerating all cases (P5) or geometry problems where visual reasoning is helpful (P14).
> Second, for harder proof-based competitions, we have already provided such an error analysis with actionable insights in our prior work [3] and recently performed a qualitative analysis on our evaluation of the IMO, the results of which we will include in the final revision of this paper. Generally, we believe that an error analysis should provide actionable insights, which requires manual investigation in the error modes of current frontier models. While automated analyses, such as failure mode categorization, can provide a limited sense of where models often go wrong, it does not tell us how to improve these mistakes beyond very basic insights. In contrast, manual human analysis can provide much deeper insights. For instance, in our analysis of the IMO 2025, we found that Gemini-2.5-Pro provides bogus citations when failing to solve a problem. Furthermore, Grok-4 often produces proofs that simply state the final answer. This can be mitigated by training the model with more varied prompts, preventing its prompt-sensitivity. Since our judges already perform a detailed analysis of answers while grading, this detailed analysis can be easily provided for graded competitions, but not so easily for automated competitions.
>
> **Additional comments:**
> Finally, we want to highlight our continued effort over the past few months to keep our dynamic benchmark up to date, as promised in the paper. Specifically, we’ve added the IMO 2025 as a new proof-based benchmark, included the latest Project Euler problems to test mathematical coding skills on more complex challenges, and added another final-answer competition, CMIMC 2025.
>
> [1] TechCrunch. Meta’s vanilla Maverick AI model ranks below rivals on a popular chat benchmark. Kyle Wiggers, April 11, 2025.
>
> [2] Lemmata. AI Did Well on the 2025 AIME. Greg Burnham, February 11, 2025.
>
> [3] Petrov, Ivo, et al. "Proof or bluff? evaluating llms on 2025 usa math olympiad." arXiv preprint arXiv:2503.21934 (2025).

---

> > ### Comment · Reviewer_smDA · 2025-08-05
> >
> > Thank you to the authors for their responses.
> >
> > Regarding the potential contamination issue, I suggest incorporating the relevant discussion into the revised manuscript. For the failure case analysis, it would also be beneficial to include a more detailed discussion in the main content, as this can help the research community further improve the math and general reasoning performance of LLMs.
> >
> > After reading the authors' responses and the comments from other reviewers, I will keep my score.

---

### Official Review · Reviewer_DoTC · 2025-07-03

**Rating:** 5
**Confidence:** 3

**Summary:**

This paper introduces a dynamic and contamination-free benchmark for evaluating large language models (LLMs) on mathematical reasoning using newly released problems from reputable math competitions such as AIME, HMMT, USAMO, and SMT. Unlike prior benchmarks prone to training data leakage or limited to final answers, MathArena includes both final-answer and proof-based tasks—making it the first benchmark to rigorously assess natural-language proof generation.

**Dataset Code Accessibility:**

Yes

**Ethical Considerations:**

No, there are no or only very minor ethics concerns

**Limitations Weaknesses:**

Potential for Future Leakage: Despite careful design, some new competition problems still appeared online in similar forms, posing a residual risk of indirect contamination. Are there any designs for randomly generating new math problems?

**Strengths Contributions:**

1. MathArena ensures an uncontaminated benchmark—addressing a major flaw in many prior evaluations.


5. The paper provides detailed statistical evaluation, confidence intervals, cost-performance tradeoffs, and per-domain accuracy—making its findings robust and actionable.

6. By aligning with existing math competitions, MathArena naturally reflects the kind of problems LLMs may face in real educational or scientific use cases.

---

> ### Author Rebuttal · Authors · 2025-07-30
>
> Thank you for your review, we provide our response below.
>
>
> **Q1: Despite careful design, some new competition problems still appeared online in similar forms, posing a residual risk of indirect contamination. Are there any designs for randomly generating new math problems?**
>
> First, we want to highlight that the observed contamination was much more pronounced in AIME than HMMT. This makes sense because AIME is only a first selection test that is more accessible to “beginner” students. However, all our other competitions are on the level of HMMT or higher, and more difficult problems are less likely to appear online, resulting in the smaller contamination. For the most difficult competitions such as the USAMO and IMO, contamination is incredibly unlikely to happen due to the high stakes involved and the extreme rigour applied when creating these problems.
> Automatically generated problems are much more likely to contain errors and do not meet the high standards for problems created for these competitions. Furthermore, one would need to address several difficult challenges to generate these problems. For instance, the generated problems need to be difficult enough to test models on the same level as the problem generator. While we are currently looking into how to resolve these challenges, we consider this out-of-scope for our current work.
>
> **Additional comments:**
> Finally, we want to highlight our continued effort over the past few months to keep our dynamic benchmark up to date, as promised in the paper. Specifically, we’ve added the IMO 2025 as a new proof-based benchmark, included the latest Project Euler problems to test mathematical coding skills on more complex challenges, and added another final-answer competition, CMIMC 2025.

---

### Decision · Program_Chairs · 2025-09-18

**Decision:**

Accept (poster)

**Comment:**

This paper unanimously receives positive rates thanks to the dataset significance for evaluating LLMs’ reasoning-based proving capability. Although overall reviews are positive, the clarification in the rebuttal should be reflected in the final draft, including data missing citation, scalability issues, and most importantly data privacy and copyright concerns.